# ENGIBENCH: A BENCHMARK FOR EVALUATING LARGE LANGUAGE MODELS ON ENGINEERING PROBLEM SOLVING

## ABSTRACT

Large language models (LLMs) have shown strong performance on mathematical reasoning under well-posed conditions. However, real-world engineering problems require more than mathematical symbolic computation—they need to deal with uncertainty, context, and open-ended scenarios. Existing benchmarks fail to capture these complexities. We introduce EngiBench, a hierarchical benchmark designed to evaluate LLMs on solving engineering problems. It spans three levels of increasing difficulty (foundational knowledge retrieval, multi-step contextual reasoning, and open-ended modeling) and covers diverse engineering subfields. To facilitate a deeper understanding of model performance, we systematically rewrite each problem into three controlled variants (perturbed, knowledge-enhanced, and math abstraction), enabling us to separately evaluate the model's robustness, domain-specific knowledge, and mathematical reasoning abilities. Experiment results reveal a clear performance gap across levels: models struggle more as tasks get harder, perform worse when problems are slightly changed, and fall far behind human experts on the high-level engineering tasks. These findings reveal that current LLMs still lack the high-level reasoning needed for real-world engineering, highlighting the need for future models with deeper and more reliable problem-solving capabilities. Our source code and data are available at https://anonymous.4open.science/r/EngiBench-05DF.

## 1 INTRODUCTION

Large language models (LLMs) have demonstrated promising capabilities in a range of mathematical reasoning tasks, from foundational skills such as basic computation and structured problem-solving (Cobbe et al., 2021), multi-step reasoning (Shao et al., 2024; Wei et al., 2022), to more complex applications like mathematical modeling (Guo et al., 2025) and the generation or verification of mathematical proofs (Yang et al., 2023; Lin et al., 2025; Ren et al., 2025). However, just using mathematical reasoning is not enough for real-world applications. In practice, many high-impact use cases occur not in abstract mathematical domains, but in engineering contexts, where problems are grounded in physical systems and require balancing uncertainty and constraints inherent to real-world decision making. These characteristics require not only mathematical computation, but also need broader capabilities to understand engineering contexts and solve complex engineering problems.

Engineering problems differ fundamentally from mathematical problems. Mathematical problems aim for abstract theoretical rigor and universality, and are typically characterized by complete information within a clearly defined problem space (Hendrycks et al., 2021). In contrast, engineering problems are driven by the need to find "good enough" and feasible solutions for specific objectives, which are often open-ended, highly context-dependent, and must be achieved within real-world constraints (Dym et al., 2005). For example, designing a drone system (Table 1) requires identifying relevant operational requirements and balancing objectives such as range, payload, and energy limits, illustrating the practical complexity that characterizes real-world engineering tasks. As illustrated in Figure 1, solving real-world engineering problems requires more than retrieving a formula or executing a single calculation. It involves a sequence of interdependent cognitive steps that span from understanding the problem context to formulating robust, feasible solutions. We define this broader set of competencies as the engineering problem-solving capability, comprising four interconnected

Figure 1: Task taxonomy of EngiBench organized by difficulty, capability, and subfield. Problems are grouped into three difficulty levels, with Level 3 specifically designed to evaluate engineering problem-solving capabilities. All tasks are additionally categorised into three major engineering subfields.

dimensions: *information extraction, domain-specific reasoning, multi-objective decision-making, and uncertainty handling*.

Despite the broader requirements of real-world engineering tasks, most existing benchmarks focus narrowly on well-defined mathematical problems. Benchmarks such as GSM8K (Cobbe et al., 2021), MATH (Hendrycks et al., 2021), and Omni-MATH (Gao et al., 2024a) primarily assess symbolic reasoning, calculation, and formal problem-solving under clean and fully specified conditions. While these benchmarks have driven progress in mathematical reasoning, their support for engineering tasks remains unclear. Although some include basic engineering questions, they fail to capture the deeper reasoning required for real-world problem solving (Hendrycks et al., 2021; Wang et al.; Albalak et al., 2025; Du et al., 2025). In addition, these benchmarks rely on publicly available datasets without rewriting that may overlap with LLM pretraining corpora, raising concerns about benchmark contamination and overclaimed performance (Deng et al., 2024; Huang et al., 2025; Sainz et al., 2023). For example, GSM1k introduces human-written problems in the style of GSM8k to avoid data overlap, revealing up to 8% performance drops and potential overfitting (Zhang et al., 2024). Without proper safeguards, evaluations may measure memorization rather than true generalization, particularly in engineering contexts requiring practical reasoning. Solely using unmodified public questions thus inadequately assesses true engineering capabilities, limiting insights into real-world model performance.

In this work, we introduce an evaluation framework designed to systematically assess LLMs on engineering problem-solving. It spans a wide range of engineering subfields. Meanwhile, **EngiBench** is designed around the broader concept of *engineering problem-solving capability*, evaluating LLMs across multiple dimensions aligned with the demands of practical engineering contexts. As illustrated in Figure 1, it consists of three progressively task levels. We use a structured data construction strategy consisting of three key aspects. First, we systematically rewrite public questions through numerical and semantic perturbations to minimize overlap with pretraining datasets. Second, we introduce controlled problem variations, including knowledge-enhanced and math abstraction versions, to enable fine-grained analysis of model capabilities. Finally, we adopt rubric-based evaluation for open-ended tasks, using expert-designed scoring criteria to assess model performance across key engineering problem-solving capabilities. Together, these measures yield a diverse and high-quality dataset that supports rigorous and contamination-limited evaluation of LLMs' engineering problem-solving abilities.

Experiment results show that our benchmark reveals clear performance stratification across difficulty levels, with higher-level tasks exposing distinct capability gaps. In addition, our perturbed version induce performance drops, even in strong models, revealing that prior evaluations may overestimate true generalization. Most critically, current LLMs consistently underperform on Level 3 tasks involving open-ended, high-level engineering reasoning, falling well short of human expert performance. These results suggest that today's LLMs remain far from reliably solving real-world engineering problems, leaving substantial room for future research.

Our contributions can be summarized as follows: (1) We are among the first to systematically evaluate LLMs on real-world engineering problems; (2) We design a hierarchical benchmark with three difficulty levels and multiple problem variants, enabling fine-grained analysis of model reasoning capabilities and limitations; (3) Unlike prior benchmarks, our benchmark systematically evaluate LLM performance on open-ended engineering tasks; (4) We evaluate a broad set of mainstream LLMs, providing insights that can aid future model development and enhance engineering capabilities.

## 2 RELATED WORKS

**LLMs for Engineering Problems.** LLMs possess logical-reasoning skills, domain knowledge, and the capacity for multi-step inference that surpass earlier AI paradigms, making them promising tools for tackling complex challenges. Engineering centers on understanding complex problems, building mathematical models, and discovering feasible solutions, making it highly relevant to real-world challenges and a critical domain for evaluating advanced reasoning capabilities. Although LLMs are increasingly applied to simulation, modeling, and system design, their true proficiency in engineering problem solving remains unclear because current benchmarks are inadequate (Wang et al., 2024b; Ma et al., 2024; Tang et al., 2024; Cheng et al., 2025). Some general-purpose benchmarks – like MMLU (Hendrycks et al., 2021), MMLU-Pro (Wang et al.), BIG-Math (Albalak et al., 2025), and SuperGPQA (Du et al., 2025) – include a few engineering-flavoured questions, but these are mostly fact-recall multiple-choice items that ignore authentic engineering reasoning. Domain-specific benchmarks do exist, such as EEE-Bench (Li et al., 2024), ElecBench (Zhou et al., 2024), FEABench (Mudur et al., 2025), TransportBench (Syed et al., 2024), and JEEBench (Arora et al., 2023). However, these benchmarks typically focus on single disciplines and closed-ended tasks, providing limited support for evaluating open-ended and cross-disciplinary engineering reasoning. Moreover, none of these efforts are explicitly designed to evaluate key engineering problem-solving capabilities. We introduce a multi-level engineering benchmark spanning multiple subfields that emphasizes not only closed-form tasks but also open-ended problems, enabling a more comprehensive evaluation of the essential skills needed for effective real-world engineering decision-making.

**LLM for Mathematical Problems.** A closely related area that has been extensively studied is mathematics. Because solving mathematic problems demands strong logical ability, multi-step reasoning, and symbolic manipulation, it has become a primary proving ground for evaluating LLMs. Early benchmarks focus on elementary problems (Cobbe et al., 2021; Hendrycks et al., 2021; Patel et al., 2021; Amini et al., 2019) and higher-level symbolic reasoning (Hendrycks et al., 2021; Albalak et al., 2025). Recent efforts like MiniF2F (Zheng et al., 2022), UniMath (Liang et al., 2023), Omni-MATH (Gao et al., 2024b), and MathVista (Lu et al., 2024) expand to theorem proving and multimodal tasks. MATH-Vision (Wang et al., 2024a) improves coverage by introducing diverse topics and difficulty levels from real competitions, and SMART-840 (Cherian et al., 2024) benchmarks model performance against human children across grades. While these benchmarks provide rigorous evaluations of mathematical competence, they do not capture engineering-specific reasoning such as modeling, decision-making under constraints, or domain-based assumptions. Our work builds on their methodological insights but shifts the focus toward real-world engineering tasks.

**Evaluation Challenges.** Evaluating the capability of LLMs to solve engineering problems is challenging due to the inherent complexity involved. Current evaluation methods for LLMs fall into four main categories: reference-based, task-oriented, preference-based, and rubric-based. The first two are effective for problems with clear ground truths or executable outputs – e.g., MathVista (Lu et al., 2024), CHAMP (Mao et al., 2024) (reference-based), and EEE-Bench (Li et al., 2024), FEABench (task-oriented) (Mudur et al., 2025). However, the core capabilities of the engineering field we are discussing cannot be effectively evaluated by such closed-form problems. For open-ended tasks, preference-based methods such as MT-Bench-101 (Bai et al., 2024) use pairwise comparisons, but are often biased by model-specific generation patterns, limiting objectivity and real-world applicability. Rubric-based evaluations aim to improve transparency by scoring along multiple criteria, with general-purpose frameworks like Prometheus (Kim et al., 2024) focusing on abilities such as context retention and rephrasing.

## 3 METHODOLOGY

### 3.1 ENGINEERING PROBLEM-SOLVING CAPABILITY

Engineering problems typically require practical, context-aware solutions under real-world constraints (Dym et al., 2005), fundamentally differing from mathematical problems that emphasize well-defined, closed-form problem spaces (Hendrycks et al., 2021). While both fields value abstraction and logical rigor, engineering problem-solving involves interconnected cognitive steps, from understanding problem context to formulating robust, feasible solutions (see Figure 1 and Table 1). We define this as *engineering problem-solving ability*, comprising four key dimensions: *information extraction, domain-specific reasoning, multi-objective decision-making, and uncertainty handling*. These dimensions reflect well-established paradigms in engineering modeling, including information filtering, multiobjective and constraint-based formulation, and uncertainty and robustness analysis.

Table 1: Hierarchical difficulty from mathematics to real-world engineering. This illustrates three levels of increasing complexity. Examples show the progression from closed-form math problems to open-ended engineering scenarios.

| Level | Definition | Example |
|---|---|---|
| Mathematics | Mathematical tasks are typically well-posed and self-contained, with complete information and clearly defined solution spaces. | A machine produces 45 parts per minute. If it operates continuously for 2 hours, how many parts will it produce in total? ☞ *This task requires only basic multiplication and does not involve any domain knowledge. It represents a typical closed-form numerical computation problem.* |
| 💡 **Upgrading Condition:** Incorporating domain-specific engineering knowledge | | |
| Engineering Level 1: Foundational Knowledge Retrieval | Apply basic engineering concepts or formulas to structured problems via single-step computation. | A drone operates at a constant power of 200W for 30 minutes. Calculate the total energy consumption in joules. ☞ *This task requires applying the basic physical formula $E = P \times t$, with unit conversion from minutes to seconds. It tests the model's ability to retrieve and apply foundational engineering knowledge in a single-step calculation.* |
| 💡 **Upgrading Condition:** Multi-step reasoning and contextual integration | | |
| Engineering Level 2: Contextual Reasoning | Perform multi-step reasoning under well-defined constraints by integrating conditions and domain knowledge. | A drone needs to fly 6 km. The first half is uphill, increasing power usage by 20%, while the second half is flat at 180W. The drone flies at 30 km/h and uses a battery rated at 8000mAh, 11.1V. Can the battery support the trip? ☞ *This task requires multi-step reasoning: estimating flight time, adjusting power consumption, and comparing with battery capacity.* |
| 💡 **Upgrading Condition:** Solving open-ended, under-specified problems | | |
| Engineering Level 3: Open-ended Modeling | Solve open-ended, real-world problems through information extraction, trade-off reasoning, and uncertainty handling. | Design a drone system for urban delivery that balances multiple factors, including flight range, payload capacity, and cost control. Propose a feasible solution and justify your design decisions. ☞ *This is an open-ended problem with incomplete constraints and potentially conflicting objectives, requiring information extraction, trade-off analysis, and robustness under uncertainty.* |
| 🔍 Information Extraction | Identify and extract relevant information from complex or redundant problem descriptions. | Identify critical variables—such as payload weight, wind speed, flight duration, and battery margin—from complex or verbose task descriptions. |
| 📖 Domain-specific Reasoning | Apply specialized engineering principles and structured knowledge to guide logical inference and solution formulation. | Apply specialized engineering knowledge—such as flight mechanics and battery discharge principles—to formulate models and perform technical analysis. |
| ⚖️ Multi-objective Decision-making | Make justified trade-offs between competing in the absence of a single optimal solution. | Justify trade-offs among competing objectives like range, cost, safety, and operational efficiency when no single optimal solution exists. |
| 🌦️ Uncertainty Handling | Ensure solution robustness by reasoning under incomplete, variable, or ambiguous real-world conditions. | Account for unpredictable factors such as weather, task variation, and battery aging, and design robust strategies (e.g., adding 20% battery reserve) to ensure reliable performance. |

- *Information extraction* refers to the ability to identify and retrieve critical information from complex or redundant problem descriptions. It involves recognizing relevant variables, constraints, and objectives while distinguishing them from irrelevant or distracting details. This capability reflects the model's proficiency in processing unstructured inputs and converting them into structured representations that facilitate subsequent reasoning. Its significance lies in its capacity to accurately capture the essential elements of a problem, thereby minimizing errors in subsequent reasoning processes and ultimately enabling the precise formulation of effective and implementable solutions.

- *Domain-specific reasoning* refers to the model's ability to apply specialized engineering knowledge such as physical principles, empirical rules, and practical engineering conventions to interpret a given scenario and formulate appropriate solutions. This includes understanding when certain approximations are valid, recognizing implicit assumptions commonly made in specific domains, and selecting solution strategies that align with real-world engineering practices. Such reasoning requires both conceptual understanding and practical judgment, distinguishing engineering tasks from purely mathematical problem solving.

- *Multi-objective decision-making* denotes the capability to evaluate and balance competing objectives in situations where no single optimal solution exists. Engineering problems commonly involve trade-offs among factors such as cost, performance, and safety. This dimension reflects the model's ability to navigate such trade-off spaces and justify rational decisions within given constraints. Consequently, it is this inherent requirement for trade-offs that imparts engineering problems with their distinctive characteristics of multiplicity, openness, and flexibility compared to traditionally studied problem domains.

- *Uncertainty handling* characterizes the capability to reason under conditions of incomplete or variable information. Real-world engineering scenarios frequently involve missing data, noisy inputs, or dynamic conditions. This dimension evaluates whether a model can anticipate such uncertainties, incorporate safety margins or adaptive strategies, and consistently deliver robust and reliable solutions despite these challenges. Effectively managing uncertain and ambiguous information, including making informed assumptions or estimations, is thus a critical yet complex challenge that LLMs must address to successfully solve practical engineering problems.

### 3.2 PROBLEM HIERARCHICAL DIFFICULTY DESIGN

As discussed above, the capabilities involved in solving engineering problems are multifaceted and complex, making it challenging to evaluate them comprehensively through any single task. Each capability emphasizes distinct cognitive demands and cannot be adequately represented within a single hierarchical dimension. Without a clear taxonomy for engineering problem-solving, it is difficult to pinpoint the specific skills in which a model may be deficient. To address this issue, we introduce a structured evaluation framework. Unlike previous benchmarks that merely aggregate

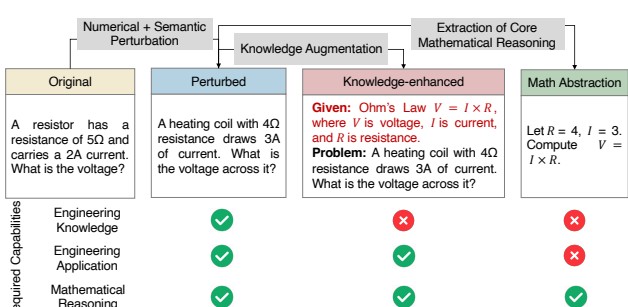

Figure 2: We create variants of the original problem to test different reasoning skills. *Perturbed* changes context and numbers to assess robustness. *Knowledge-enhanced* adds domain knowledge to focus on reasoning. *Math Abstraction* isolates engineering knowledge to test math ability. Each version targets specific capabilities.

tasks, our framework classifies tasks according to the core capabilities required by engineering scenarios. As illustrated in Table 1, engineering problem-solving spans three levels: foundational knowledge retrieval, contextual reasoning, and open-ended modeling. This hierarchy mirrors the cognitive progression in engineering problem-solving—from applying basic formulas to reasoning under uncertainty and conflicting objectives. EngiBench reflects this hierarchical organization through a three-level difficulty framework.

1. **Level 1.** Tasks are well-structured and self-contained, typically requiring the model to apply fundamental engineering formulas in a single step. They emphasize factual recall, precise computation, and minimal contextual reasoning. This level evaluates whether the model possesses a reliable engineering knowledge base and can consistently retrieve and apply it for straightforward problems.

2. **Level 2.** Tasks extend beyond formula application to multi-step reasoning under explicit contextual constraints such as units, physical limits, or interdependent variables. Problems remain well-specified and have unique answers, but require interpreting structured descriptions and integrating domain knowledge across steps. Crucially, unlike Level 1, simple recall is insufficient—models must handle structured complexity to arrive at the correct solution.

3. **Level 3.** Tasks mirror real-world challenges by being under-specified, ambiguous, or involving conflicting objectives. They require advanced reasoning across four dimensions: information extraction, domain-specific reasoning, multi-objective decision-making, and uncertainty handling. Unlike Levels 1 and 2, problems lack unique correct solutions, and evaluation focuses on how well models demonstrate robust and adaptive reasoning under open-ended conditions.

## 3.3 DATASET CONSTRUCTION

**Data Sources.** We collect data from three primary sources: problems selected from existing public benchmarks, university educational materials, and modeling competitions. These problems reflect the intended hierarchy of difficulty described above and address the lack of open-ended engineering modeling problems with expert-defined evaluation criteria in existing datasets.

**Construction Process.** Levels 1 and 2 consist of structured problems with standard answers, extracted from benchmarks such as SuperGPQA (Du et al., 2025), MMLU (Hendrycks et al., 2021), MATH (Hendrycks et al., 2021), GSM8k (Cobbe et al., 2021), Orca-Math (Mitra et al., 2024), HARP (Yue et al., 2024), Omni-MATH (Gao et al., 2024b), Big-Math (Albalak et al., 2025), and selected university resources. All problems were standardized and validated. Level 3 introduces the first systematic collection of open-ended engineering tasks, with 43 problems curated from modeling competitions, each accompanied by official scoring rubrics and reference solutions from top performers. Problems were carefully reformatted to ensure clarity and evaluability by LLMs.

**Annotation and Quality Control.** Level 3 was annotated by 20 PhD students and engineering professionals, supported by GPT-4.1 and Gemini 2.5 Flash as auxiliary tools. Detailed scoring guidelines ensured consistency and fairness, and inter-annotator agreement was high. From nearly 1,000 competition problems, only those with official rubrics were retained, with extensive reformatting of formulas, tables, and diagrams. Automated scoring scripts are released with the dataset, enabling reproducible evaluation closely aligned with human ratings.

**Coverage and Classification.** EngiBench spans three subfields: Systems & Control (916 problems), Physical & Structural (334 problems), and Chemical & Biological (467 problems). This categorization reflects differences in problem focus, required knowledge, and reasoning approaches.

### 3.4 Controlled Problem Variations for Fine-grained Capability Analysis

Multiple factors may influence LLMs performance on individual problems. Potential reasons include insufficient domain-specific knowledge, calculation errors, or difficulties in accurately interpreting the engineering context. Merely collecting problems to test overall accuracy provides only a broad indication of comprehensive performance. We propose to systematically rewrite problems to conduct controlled experiments, enabling more fine-grained analyses of LLM performance on our benchmark. This approach allows us to isolate particular challenges, evaluate the robustness of model, and detect potential data leakage issues (Huang et al., 2025; Zhang et al., 2024; Mirzadeh et al., 2025; Srivastava et al., 2024; Gulati et al., 2024). By comparing model performance across different problem variations, we gain deeper insights into the specific capabilities required for realistic engineering tasks.

Motivated by this, we design three controlled variations for each problem, each targeting a distinct aspect of the reasoning process, as illustrated in Figure 2. Unlike prior robustness benchmarks, these versions are explicitly constructed for engineering scenarios: the *knowledge-enhanced* and *math abstraction* versions are, to our knowledge, the first systematic variants tailored to diagnose domain-specific reasoning failures rather than general robustness. This design enables fine-grained capability analysis by isolating knowledge gaps, contextual dependencies, and mathematical reasoning skills. The detailed construction procedure is provided in the Appendix. Starting from an *original version*, we construct the following three versions:

(1) The *perturbed version* introduces numerical and semantic perturbations to the original problem, thereby reducing overlap with pretraining datasets. (2) The *knowledge-enhanced version*, built upon the perturbed version, explicitly provides relevant engineering knowledge, such as formulas, physical constants, and domain-specific definitions. This version helps to diagnose whether model errors stem specifically from a lack of critical knowledge. (3) The *math abstraction version* removes all contextual and domain-specific elements, reformulating the problem purely as a symbolic computation task. This isolates the model from the engineering context, reverting the evaluation to well-established mathematical reasoning and computational capabilities. Consequently, this version explicitly illustrates the impact of the engineering context on model performance.

These three variations, along with the original versions, are constructed systematically for all tasks in Level 1 and Level 2. For Level 3 tasks, however, the open-ended and inherent complexity typically render knowledge enhancement and mathematical abstraction impractical. Hence, only the original and perturbed versions are provided for this level. Moreover, we provide a detailed scoring criteria for Level 3, based on the official evaluation criteria disclosed by the competition organizers. This rubric enables assessment of an LLM's response to the specific requirements of each capability dimension.

## 4 Experiments

### 4.1 Experiment Setup

**Evaluated LLMs.** As the first batch, 16 LLMs were evaluated under the zero-shot setting, covering a representative range of model types. Specifically, we include: (1) closed-source models such as GPT-4.1, GPT-4.1 Mini, and GPT-4.1 Nano from OpenAI (Achiam et al., 2023); Claude 3.7 Sonnet and Claude 3.5 Sonnet from Anthropic (Anthropic, 2024a;b); and Gemini 2.5 Flash and Gemini 2.0 Flash from Google DeepMind (Team et al., 2023; 2024); (2) open-source models, including GLM-4-32B and GLM-4-9B from THUDM (GLM et al., 2024), Qwen2.5-72B and Qwen2.5-7B from Alibaba (Yang et al., 2024), Llama 4 Maverick (referred to as Llama 4) and Llama 3.3-70B (referred to as Llama 3.3) from Meta (Grattafiori et al., 2024), and DeepSeek-V3-671B (referred to as DeepSeek-V3) and DeepSeek-R1-Distill-Qwen-1.5B (referred to as DeepSeek-R1 7B) from DeepSeek (Liu et al., 2024; Guo et al., 2025), Mixtral-8x7B-Instruct-v0.1 (referred to as Mixtral 8x7B) from Mistral AI (Jiang et al., 2024). This selection spans diverse model sizes, training paradigms, and accessibility levels. We ensured consistent formatting and output parsing across all models.

**Evaluation protocols.** A key challenge in evaluating engineering problem-solving lies in determining not whether a solution is correct, but whether it is good enough given practical constraints. Unlike mathematical problems with definitive answers, real-world engineering tasks often involve uncertainty, redundant information, and competing objectives. These characteristics make binary judgments insufficient for capturing the quality and completeness of a solution.

For Level 1 and Level 2, which consist of well-structured problems with clear solutions, we adopt binary scoring. A response is marked correct only if it exactly matches the reference answer, and

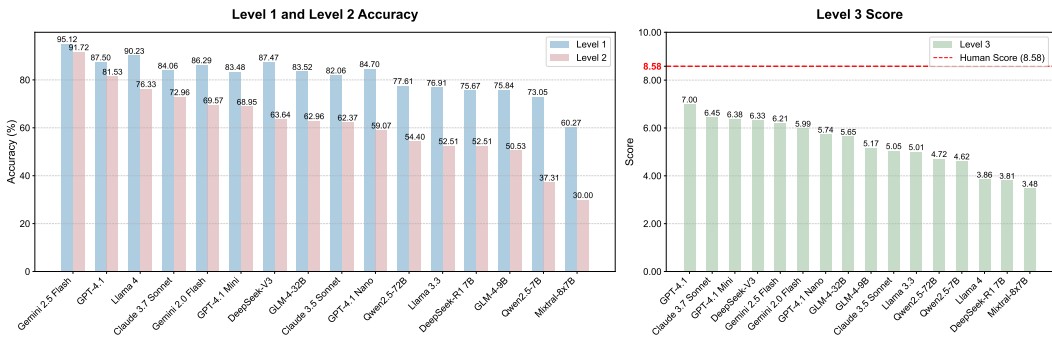

Figure 3: Overview of model performance across engineering reasoning tasks. The left subfigure shows model accuracy on Level 1 and Level 2 tasks, while the right subfigure presents expert-assigned scores on Level 3 open-ended tasks, with the human expert score indicated by the red line.

overall performance is reported as accuracy. Level 3 tasks are open-ended and under-specified, lacking a single correct answer, which makes it essential to evaluate not just correctness but the quality and completeness of a model's reasoning. Their evaluation depends on how well a model extracts relevant information, applies domain knowledge, balances competing objectives, and reasons under uncertainty. We therefore employ a rubric-based scoring framework, constructed from officially released and expert-designed criteria and refined with LLM assistance. For each problem, we extract the rubric points relevant to our target competencies and convert them into concrete scoring items. To ensure scoring quality, all results were reviewed by PhD-level professionals with expertise in mathematical modeling.

Also, we introduce human scores for Level 3 tasks for comparison with LLMs' performance. We obtain human scores from two sources: award-winning competition submissions (original version) and manual solutions by top-performing students for the perturbed version. All responses are evaluated using the same rubric as LLM outputs to ensure consistency and fairness.

## 4.2 RESULTS

### 4.2.1 OVERALL

**Model stratification and design validation.** Model performance exhibits a clear downward trend from Level 1 to Level 3, demonstrating the effectiveness of our hierarchical difficulty design. As shown in Figure 3, most models achieve high accuracy on Level 1, perform moderately on Level 2, and struggle significantly on Level 3. This progression indicates that our hierarchical framework successfully separates problems by cognitive difficulty, with each level revealing distinct capability thresholds. The results validate that a multi-level design is necessary to capture the full spectrum of engineering problem-solving capabilities.

**Evaluating high-level engineering reasoning.** Level 3 is designed to assess high-level engineering reasoning that goes beyond formulaic computation. Unlike Level 1 and Level 2, which focus on structured problem solving, Level 3 features open-ended and underspecified tasks that better reflect real-world engineering challenges. The sharp performance drop at this level reveals the current limitations of LLMs in handling such complex scenarios. Besides, the gap between LLMs and human experts at Level 3 also reveals a key deficiency in high-level engineering capabilities. All evaluated models score well below the human expert, who achieves an average of 8.58, indicating that current LLMs are still far from reliably handling complex engineering problems. This underscores the need for further research to bridge this gap.

**Smaller-scale LLMs struggle with complex tasks.** While all LLMs show room for improvement on complex, open-ended engineering tasks, smaller-scale LLMs exhibit significantly greater limitations. As task complexity increases, performance disparities widen. At Level 1, most models still cluster within 70–90%. But at Level 2, leading models such as GPT-4.1 and Gemini 2.5 Flash achieve accuracies above 80%, whereas DeepSeek-R1 7B reaches only about 52% and other lightweight models often fall below 40%. This divergence is most pronounced at Level 3, where state-of-the-art models approach scores of 7.0, while lightweight models remain under 4.0. These results indicate that EngiBench is far from saturated and continues to provide meaningful differentiation across scales.

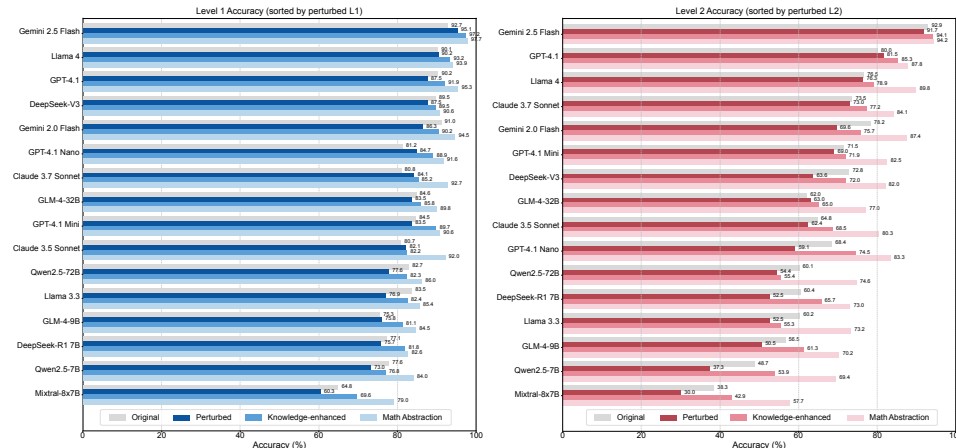

Figure 4: Accuracy of LLMs on Level 1 (left) and Level 2 (right) tasks across the original setting and three variants: Original, Perturbed, Knowledge-enhanced, and Math Abstraction. Drops in the Perturbed version indicate sensitivity to input changes, while gains in the latter two show that current LLMs require external knowledge or reformulation to improve accuracy—highlighting their lack of these abilities.

**Robustness and contamination risk.** Some LLMs may achieve high scores not through internal reasoning, but due to overlap with pretraining data. To reveal this, we introduce perturbed variants that modify surface details but keep the core structure unchanged. As shown in Figure 4, performance remains relatively stable on Level 1 but drops sharply on Level 2—e.g., 9.3% for GPT-4.1 Nano, 11.4% for Qwen2.5-7B, and 8.3% for Mixtral-8x7B. These declines reveal that many models rely on superficial pattern matching rather than robust reasoning. This underscores the value of perturbation-based evaluation in exposing overestimated capabilities and assessing true generalization.

### 4.2.2 PERFORMANCE FOR LEVEL 1 & LEVEL 2 TASKS

Our results show that adding explicit domain knowledge significantly improves model accuracy across all levels, especially for weaker models. As shown in Figure 4, models perform consistently better on knowledge-enhanced variants than on perturbed inputs. These gains may reflect two common failure modes: either the model lacks sufficient domain knowledge, or it fails to recognize when and how to apply it during multi-step reasoning. The use of explicit knowledge prompts thus provides a useful diagnostic signal for distinguishing between knowledge gaps and reasoning failures—an important capability dimension for engineering benchmarks.

In addition, LLMs' performance further improves when problems are abstracted into symbolic mathematical form, eliminating engineering context. As shown in Figure 4, most models achieve their highest accuracy under this variant, particularly smaller-scale LLMs that struggle with contextual interpretation. This trend reveals that the primary difficulty in engineering problem-solving lies not in the computation itself, but in the upstream reasoning required to structure the problem from natural input. This affirms the necessity of assessing reasoning steps that precede formula application—steps often overlooked by traditional math benchmarks.

Smaller-scale LLMs exhibit significantly greater performance variation across different input versions, revealing limited generalization and unstable reasoning processes. As shown in Figure 4, Qwen2.5-7B drops by 11.4% under the perturbed version, but gains 16.6% when explicit domain knowledge is added and a further 15.5% under math abstraction. In contrast, Gemini 2.5 Flash—a top-performing model—remains largely stable, with only minimal changes relative to its perturbed performance (-1.2%, +2.4%, and +0.1%). This contrast highlights that smaller-scale models are sensitive to input formulation and often rely on surface patterns rather than consistent, context-aware reasoning.

### 4.2.3 PERFORMANCE FOR LEVEL 3 TASKS

**Dimension-wise and model-wise performance.** As shown in Figure 5a, human experts lead across all four dimensions with a balanced capability profile. In contrast, LLMs show uneven performances: they perform best on redundant information extraction, moderately on multi-objective decision-making, and poorly on domain-specific reasoning and uncertainty handling—highlighting a lack of deep, context-aware reasoning. Results also demonstrate that model performance also correlates with scale and accessibility. Larger, closed-source models like GPT-4.1 and Gemini 2.5 Flash consistently score above 6, demonstrating broader coverage though limited in-depth analysis. In contrast, smaller

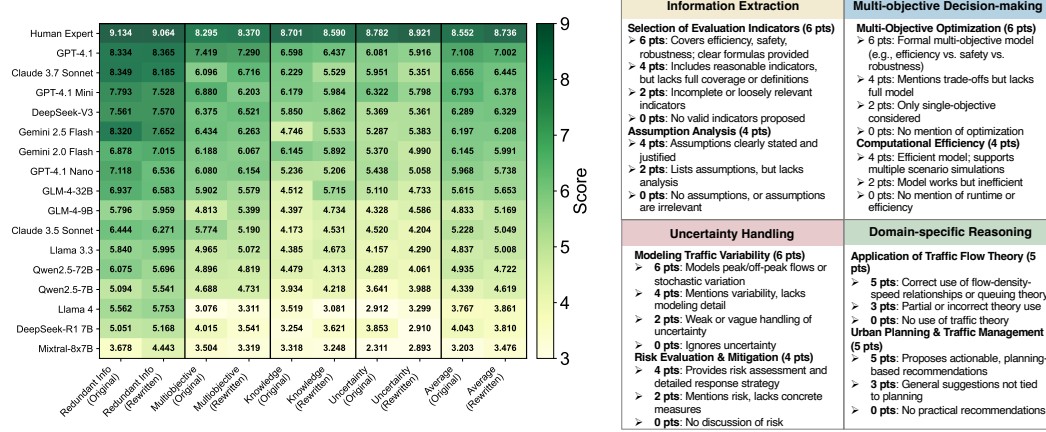

(a) Level 3 Model Evaluation.

(b) Scoring rubric example.

Figure 5: Level 3 Model Evaluation and Scoring Rubric. This figure summarizes Level 3 evaluation results and scoring standards. Subfigure (a) reports average model scores across four capabilities under both original and rewritten inputs. Subfigure (b) shows an example rubric outlining scoring criteria across capability dimensions.

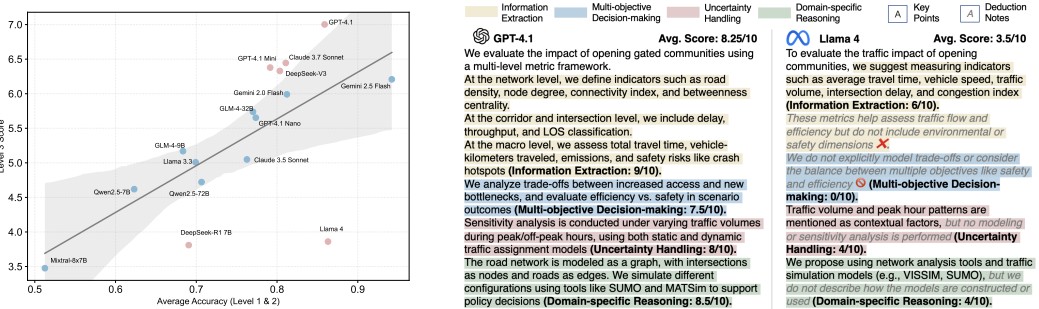

Figure 6: Correlation between structured tasks (Level 1&2) and open-ended tasks (Level 3).

Figure 7: Case study showing why Llama 4 received low Level 3 scores.

open-source models (e.g., Qwen2.5-7B, Mixtral-8x7B) average below 4, often omitting key factors such as trade-offs or uncertainty handling.

**Correlation analysis.** To quantify this trend, Figure 6 illustrates the relationship between model performance on structured tasks (Levels 1 & 2) and open-ended tasks (Level 3). Overall, we observe a clear positive correlation: models that achieve higher accuracy on structured tasks tend to also perform well on open-ended tasks, suggesting a general consistency across task types.

However, few models deviate from the general trend. For example, GPT-4.1, Claude 3.7 Sonnet, and DeepSeek-V3 outperforming expectations on Level 3 tasks—showing not just factual recall but stronger reasoning and modeling abilities. In contrast, models like Llama 4 perform pretty well on structured tasks but falter on open-ended ones, revealing weak high-level reasoning. Figure 7 illustrates this gap: Llama 4 scores 0 in multi-objective decision-making due to missing trade-off analysis, while GPT-4.1 provides a structured evaluation and scores 7.5. A similar shortfall also appears in uncertainty handling. These examples show that Llama 4 can recall facts but struggles to apply them in complex, judgment-based scenarios.

## 5 CONCLUSION

We introduce **EngiBench**, a benchmark for evaluating LLMs on engineering problem solving across increasing levels of complexity. Our results show that while current models perform well on foundational knowledge retrieval, their performance declines significantly in multi-step contextual reasoning tasks, due to both domain knowledge gaps and limited mathematical reasoning. On open-

ended modeling tasks, even the strongest models fall short of human-level performance, revealing persistent limitations in high-level reasoning, trade-off analysis, and uncertainty handling. These findings underscore the need for LLMs to move beyond pattern matching and toward deeper reasoning capabilities for real-world engineering applications.

## 6 ETHICS STATEMENT

This work introduces a benchmark for evaluating large language models on engineering tasks. The problems are derived from publicly available benchmarks, academic competitions, and educational materials. For open-ended tasks, human participants voluntarily contributed reference solutions and evaluation scores using publicly available rubric criteria, and personal information was collected only for inclusion in the acknowledgment section with explicit consent. The dataset does not contain sensitive data or enable harmful applications. The goal of EngiBench is to promote rigorous, transparent, and fair evaluation of language models in engineering contexts, and we affirm adherence to the ICLR Code of Ethics, including principles of fairness, transparency, and research integrity.

## 7 REPRODUCIBILITY STATEMENT

To ensure reproducibility, we release all resources, including the dataset, task splits, and evaluation code, in an anonymous repository: https://anonymous.4open.science/r/EngiBench-05DF/.

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

APPENDIX

# Contents

## A  THE USE OF LARGE LANGUAGE MODELS

In this work, LLMs were used in three ways: (1) grammar checking and language polishing during paper writing, (2) generating controlled problem variants in the benchmark construction process, and (3) serving as both the models under evaluation and auxiliary judges for rubric-based scoring.

## B  FUTURE WORKS

While EngiBench establishes a strong foundation for evaluating LLMs on engineering problem-solving, several avenues remain for further development and expansion:

**Scalability Across Engineering Domains.**   EngiBench currently covers three core engineering subfields—Systems & Control, Physical & Structural, and Chemical & Biological—which together span a wide range of disciplines such as Mechanical, Electrical, and Chemical/Biological Engineering. The benchmark framework is designed to be broadly applicable and adaptable across domains. In future work, we plan to expand the dataset by incorporating problems from additional engineering disciplines to further enhance data volume and subject diversity.

**Multimodal Evaluation Extensions.**   Future versions of EngiBench will introduce a dedicated multimodal subset to evaluate models on tasks involving vision-language reasoning. This will enable systematic assessment of model performance in scenarios that demand visual interpretation alongside textual understanding.

**Support for Long-Context Reasoning.**   We plan to extend the benchmark to include long-context engineering tasks by leveraging models with expanded context windows or hierarchical processing capabilities. This will allow for evaluation of more complex, information-rich tasks currently excluded due to input length limitations.

## C  LIMITATIONS

While EngiBench provides the first systematic evaluation of LLMs on real-world engineering problems—including multi-level tasks, variant-based reasoning diagnostics, and open-ended modeling—several limitations remain that we plan to address in future work.

**Multimodal Support.**   Many real-world engineering problems require interpreting visual elements such as diagrams, schematics, or structured tables. However, the current version of EngiBench excludes such tasks due to the lack of multimodal input capabilities in most existing LLMs. To ensure consistency across evaluations, we restrict all inputs to text-only formats.

**Long-Context Support.**   Some engineering tasks involve long problem descriptions or extensive tabular data that exceed the input length limits of current LLMs. To avoid unfair model truncation effects and ensure uniform evaluation settings, such problems are not included in this version of the benchmark.

**Human-in-the-loop Curation.**   Building the dataset involves substantial human effort, including problem collection, answer generation, and variant validation. This ensures data quality and alignment with engineering standards, but also reflects the significant manual effort behind the benchmark.

## D  DATASET CURATION- ADDITIONAL DETAILS

### D.1  LEVEL 1 & LEVEL 2 EXTRACTION PROCESS

To construct a high-quality and diverse dataset for Level 1 and Level 2, we systematically extract relevant tasks from a range of established public benchmarks, including MMLU (Hendrycks et al., 2021), MATH (Hendrycks et al., 2021), GSM8k (Cobbe et al., 2021), Orca-Math (Mitra et al., 2024), HARP (Yue et al., 2024), Omni-MATH (Gao et al., 2024b), Big-MATH (Albalak et al., 2025), and competition datasets such as cn_k12, Olympiads, AOPS forum, and AMC-AIME (Huang et al., 2023). In addition to these public sources, we also incorporate university-level engineering educational materials, including assignments, examinations, and instructor-provided teaching content, to further increase task diversity and real-world relevance.

To transform mathematical and logic-oriented problems into engineering-relevant evaluation tasks, we design a structured data processing pipeline that combines LLM-based analysis with human verification to ensure engineering relevance and classification accuracy. This pipeline ensures that all included problems align with real-world engineering semantics and reasoning demands, forming the basis for Level 1 and Level 2 in EngiBench.

The processing pipeline consists of the following steps:

1. **Engineering Relevance Filtering:** Each problem is evaluated for its applicability to engineering scenarios. Problems lacking domain relevance are excluded to maintain the technical integrity of the benchmark. The prompt used to determine whether a problem pertains to engineering is as follows:

```
1    """Determine if ORIGINAL problem can be solved with ONLY
     mathematical knowledge (NO engineering background):
2    - False if requires any domain-specific knowledge
3    - True if solvable through pure mathematical calculations"""
4
```

2. **Discipline and Subfield Classification:** Relevant problems are first assigned to a specific engineering discipline (e.g., Electrical, Civil, Mechanical), and then grouped into one of EngiBench's three high-level analytical subfields: Systems & Control, Physical & Structural, or Chemical & Biological. The prompt used for assigning a problem to a specific engineering discipline is as follows:

```
1    """If yes, which engineering category? (Chemical/
     Bioengineering/Geotechnical/Energy/Nuclear/Aerospace/Automotive
     /Biomedical/Civil/Control/Electrical/Industrial/Mechanical/
     Ocean/Environmental/Other) (Please try to avoid Other)
```

```
2        If not an engineering problem, return "N/A"."""
3
```

3. **Difficulty Level Assignment:** Based on the complexity of the required reasoning process, tasks are categorized into Level 1 or Level 2. Level 1 includes basic knowledge recall and single-step computation, while Level 2 involves multi-step inference, contextual understanding, and integration of structured constraints. The prompt used for classifying the difficulty level of a problem is as follows:

```
1    """Difficulty level? (Level 1/Level 2) (Please try to avoid
     unknown):
2    - Level 1: The problem can be solved by a direct retrieval of
     information or by directly substituting values into a known
     formula i.e., the shortest possible solution path. No
     chaining of intermediate steps is required. (Example: Using Ohm
     's Law, V = IR, to directly compute voltage when given current
     and resistance.)
3    - Level 2: The problem requires multi-step reasoning meaning
     that it involves chaining together several logical deductions,
     intermediate calculations, or systematic strategies beyond a
     single direct formula application. (Example: Analyzing a
     circuit to compute total resistance by first calculating
     individual branch resistances and then combining them.)"""
4
```

## D.2  Level 3 Data Collection and Processing

To construct the Level 3 dataset in **EngiBench**, we focus on real-world, open-ended engineering tasks sourced from major mathematical modeling competitions. Specifically, we collect problems from publicly accessible archives of contests such as the China Undergraduate Mathematical Contest in Modeling (CUMCM), the Mathematical Contest in Modeling / Interdisciplinary Contest in Modeling (MCM/ICM), and the Asia and Pacific Mathematical Contest in Modeling (APMCM), covering the years 2010 to 2024.

To ensure domain relevance and evaluation consistency, we apply strict filtering criteria. We retain only problems with clear engineering context and official scoring rubrics, and exclude those that depend heavily on complex diagrams or large external tables requiring multimodal input.

We standardize the selected problems using a structured pipeline that combines LLM-based processing with human oversight. This ensures language clarity, formatting consistency, and reduced risk of data contamination. The pipeline includes the following steps:

1. **Language Normalization:** Non-English problems are translated into fluent English using machine translation, while preserving the original engineering semantics.

2. **Expression Rewriting:** To minimize potential overlap with pretraining data, each problem is paraphrased by the LLM using diverse sentence structures and reasoning styles. While surface expressions are significantly altered, the core logic, numerical values, and solution paths remain unchanged. This step produces the *perturbed version* of each task, which is used to evaluate model robustness to superficial input variations.

3. **Multimodal Simplification:** For problems containing simple figures or tables, we extract and describe the essential information using plain text or LaTeX-formatted representations to support uniform text-based evaluation.

**LLM Prompt Template:** The following instruction prompt is used to guide the LLM in modifying each problem:

```
1    """Assuming you are a question expert, please translate this question
     into English. And while ensuring that the meaning of the question
     remains unchanged (preserving all logic, values, and the type of
     reasoning required), change the way the question is expressed by
     rewriting it in a way that is radically different from your regular
     logical structure, simulating the randomness of manual rewriting by
```

```
     human experts, and using as many sentence variations as possible. If
     there is a table, please convert it into a table form using LaTeX.
     For simple pictures, please describe them directly. The question is
     required to be converted into is in str format."""
2
```

To ensure the technical rigor and domain consistency of the Level 3 dataset, the entire generation and transformation process was closely supervised and iteratively revised by doctoral-level professionals with extensive expertise in engineering and mathematical modeling. These experts reviewed both the selection of source problems and the outputs produced by the language model, verifying that each task preserved the original problem's intent, accurately reflected real-world engineering reasoning, and met the standards expected in academic and professional modeling contexts.

The details of how the original contest scoring standards were mapped into EngiBench's formal scoring rubrics are described in the later subsection (see Section F.2).

### D.3 VERSION VARIANT GENERATION

To assess model robustness and isolate specific reasoning limitations, we generate three structured variants for each Level 1 and Level 2 problem: *Perturbed*, *Knowledge-Enhanced*, and *Math Abstraction*. These variants are created through LLM prompting, with manually verified outputs to ensure alignment with the original problem logic and correctness. Below, we describe the purpose and generation criteria for each variant, accompanied by illustrative prompts.

- **Perturbed Version.** This variant alters the surface form of the original problem—either through numerical or linguistic changes—while preserving its core logic and computational requirements. The purpose is to test whether model performance stems from true reasoning ability or superficial pattern matching. A rewriting suitability code (0–3) guides the type of modification to apply. The prompt used to generate the perturbed version and related content is as follows:

```
1  """
2  1. Rewriting Suitability: Determine the type (0-3):
3    - 0: Non-rewritable (use only when necessary)
4    - 1: Modify expressions only
5    - 2: Modify numerical values only
6    - 3: Modify both expressions and numerical values
7    // Note: All rewrites must maintain the original problem logic,
       engineering context, and reasoning/computational requirements
8
9  2. Rewritten Problem: Rewrite the problem according to the type of
       rewriting suitability above. Make the answer as difficult as
       possible while ensuring that the answer is correct. (Please
       rewrite the problem in a way that is radically different from
       your regular logical structure by: (1) avoiding common
       reasoning patterns in your model, (2) simulating human expert
       manual rewriting randomness, and (3) using maximum sentence
       variation.)
10   - If 0, return original problem unchanged
11   - If 1, modify expressions only
12   - If 2, modify numerical values only
13   - If 3, modify both expressions and values
14
15 3. Rewritten Solution Process: Provide step-by-step explanation
       including all reasoning, calculations and logic. Clearly state
       if answer can be obtained directly through formula substitution
       (shortest solution path without intermediate steps).
16
17 4. Rewritten Answer: Provide correct answer for rewritten problem
       (only types 2/3 may change)"""
18
```

- **Knowledge-enhanced Version.** In this version, relevant domain knowledge—such as formulas, constants, and conversions—is explicitly provided before the original question.

This allows us to evaluate whether performance deficits are due to missing knowledge or failures in application. The question itself is unchanged to isolate the impact of added context. The prompt used to generate the knowledge-enhanced version is as follows:

```
1  """Knowledge-Enhanced Version:
2  WARNING: Make sure the final numerical answer to the converted
      mathematical problem is exactly the same as the original
      problem.
3
4  Given:
5  - List all relevant formulas or principles (e.g., Ohm's Law: V = I
      * R)
6  - Include physical constants with values if they are involved (e.g
      ., g = 9.8 m/s^2)
7  - Specify unit conversions if applicable (e.g., 1 kWh = 3.6 * 10^6
      J)
8  - State any assumptions or ideal conditions if necessary (e.g.,
      assume no heat loss)
9
10 Problem:
11 Repeat the original question exactly as stated
12
13 Example:
14 Original: "Calculate voltage across 5 Ohm resistor with 2 A
      current"
15 Enhanced:
16 "Given:
17 - Ohm's Law: V = I * R
18 - Problem: Calculate voltage across 5 Ohm resistor with 2 A
      current" """
19
```

- **Math Abstraction Version.** This version reformulates the original engineering problem into a purely mathematical format by removing all domain-specific context. Variables and operations are explicitly defined to preserve the exact calculation logic. This allows us to isolate whether reasoning failure arises from contextual understanding or mathematical ability. The prompt used to generate the math abstraction version is as follows:

```
1  """Rewrite the given problem into a purely mathematical version by
      :
2
3  a. Remove all domain-specific context (e.g., chemistry, physics,
      economics).
4  b. Keep only numbers, variables, and math operations.
5  c. If domain-specific knowledge is required (e.g., reaction ratio,
       atomic mass), extract only the final numerical ratio or
      constant and include it directly.
6  d. Maintain the exact calculation logic and final answer.
7  e. Use structured symbolic language in a compact form:
8  - Introduce variables explicitly (e.g., "Let x = 2 and y = 3.")
9  - Define the calculation clearly (e.g., "Total z = min(x, y) * 2."
      )
10 - End with "Find the result."
11
12 WARNING: Make sure the final numerical answer to the converted
      mathematical problem is exactly the same as the original
      problem.
13
14 Examples:
15
16 Original: "In the reaction: Cl2 + H2 -> 2HCl, 1 mole of Cl2 reacts
       with 2 moles of H2. How many moles of HCl can be formed?"
17 converted_problem: "Let x = 1 and y = 2. They react in the ratio x
       : y : z = 1 : 1 : 2. Total product z = min(x, y) * 2. Find the
      result."
```

```
18
19 Original: "A 2m wide platform sinks 0.01m under 60kg. Estimate its
       length assuming water density = 1000 kg/m^3."
20 converted_problem: "Let x = 60 / (2 * 0.01 * 1000). Find the
       result." """
21
```

# E  DATASET URLS, LICENSE, AND HOSTING PLAN

## E.1  DATASET INSTANCE METADATA

For the EngiBench dataset, each instance corresponds to an engineering task and is stored in a structured format. Instances are categorized according to task difficulty (Level 1, 2, or 3) and are constructed with multiple versions to enable fine-grained evaluation of different capabilities. The metadata fields for each level are described below:

**Level 1 and Level 2**  Each row in the Level 1 & 2 dataset corresponds to a closed-form or structured engineering problem, and includes the following fields:

- **problem** – Original natural language problem statement.
- **answer** – Ground truth answer to the original problem.
- **subfield** – Engineering subfield to which the problem belongs (e.g., Systems & Control).
- **category** – Topic-specific classification within the subfield (e.g., Thermodynamics).
- **difficulty** – Either `Level 1` (Foundational Knowledge Retrieval) or `Level 2` (Contextual Reasoning).
- **converted_problem** – Abstract mathematical formulation of the problem.
- **converted_problem_llm_answer** – LLM-generated response to the converted problem.
- **knowledge_enhanced_problem** – Problem reformulated with explicit formulas and domain definitions.
- **rewritten_problem** – Semantically or numerically perturbed variant of the original problem.
- **rewritten_answer** – Answer to the rewritten problem.
- **rewritten_converted_problem** – Mathematical abstraction of the rewritten problem.
- **rewritten_converted_problem_llm_answer** – LLM response to the rewritten converted problem.
- **rewritten_knowledge_enhanced_problem** – Knowledge-enhanced version of the rewritten problem.

**Level 3**  Each Level 3 instance represents an open-ended modeling task and includes both the problem prompt and a rubric-based evaluation across multiple capability dimensions:

- **question_original_language** – Native language version of the open-ended task (typically Chinese).
- **question** – English translation of the open-ended modeling task.
- **question_modified** – Semantically perturbed variant of the task.
- **subquestion_original_language** – Rubric sub-criteria in the original language.
- **subquestion** – English translation of the rubric sub-criteria.
- **subquestion_modified** – Semantically perturbed variant of the sub-criteria.
- **source_detail** – Source of the modeling task (e.g., MCM, coursework).
- **official_scoring_standard_original_language** – Original rubric definition.
- **official_scoring_standard** – English translation of rubric criteria.
- **subfield** – Engineering subfield of the task.

- **category** – Domain or topic under which the task is categorized.
- **information_extraction_score** – Score for identifying relevant variables and constraints.
- **multi_objective_decision_score** – Score for resolving trade-offs across objectives.
- **uncertainty_handling_score** – Score for reasoning under ambiguity or variable inputs.
- **domain_specific_reasoning_score** – Score for applying engineering-specific logic and formulas.

# F EVALUATION DETAILS

## F.1 LEVEL 1 & LEVEL 2 EVALUATION DETAILS

Level 1 and Level 2 tasks consist of well-structured problems with clearly defined solutions. Therefore, we adopt a **binary scoring** method. Each model-generated answer is compared against a reference answer and marked as either correct (1) or incorrect (0). Final performance is reported as overall accuracy.

To improve evaluation robustness, we introduce an automated comparison prompt executed by a large language model. This prompt is carefully designed to evaluate whether the generated answer matches the reference answer based on mathematical correctness, unit validity, and reasoning soundness. For numerical questions, a tolerance of $\pm 2\%$ is allowed to account for rounding differences in complex calculations. The model is instructed to output only a Boolean result ("True" or "False") to ensure consistent scoring across all instances. The evaluation prompt used for this process is as follows:

```
"""Please analyze these two answers carefully:
Generated Answer: {generated_answer}
Standard Answer: {correct_answer}

Follow these rules for comparison:
1. For calculation-focused problems:
    - If the numerical values match, consider it correct even if units are
      missing
    - Focus on the mathematical reasoning and final numerical result
    - Check if the core calculation steps are correct
    - For complex calculations, allow  2 % tolerance in the final
     numerical result

2. For conceptual or unit-specific problems:
    - Units and their consistency must be considered
    - The complete answer including units is required

3. Consider the answer correct if:
    - The mathematical reasoning is sound
    - The final numerical value matches (within  2 % tolerance for complex
      calculations)
    - For calculation-focused problems, matching units are not mandatory

Reply only with "True" or "False". """
```

## F.2 LEVEL 3 EVALUATION DETAILS

To convert open-ended modeling problems into evaluable tasks suitable for benchmarking LLMs, we systematically transform official scoring standards into structured rubrics aligned with the four key capabilities identified in Section 3.1: **information extraction**, **domain-specific reasoning**, **multi-objective decision-making**, and **uncertainty handling**. These capabilities form the foundation of Level 3 evaluation.

To construct these scoring rubrics from the official contest-provided evaluation standards, we used an LLM to transform raw scoring descriptions into a capability-oriented rubric aligned with our four target assessment dimensions. Each contest problem was paired with its corresponding official scoring criteria, and this combined input was passed to the LLM using a carefully designed instruction

prompt. The goal was to generate specific, well-structured rubrics that are detailed enough to capture subtle distinctions in model outputs, while remaining concise and practical for use in benchmark-scale evaluations. The overall scoring workflow is illustrated in Figure 8.

In Level 3 open-ended tasks, each problem was independently annotated and cross-checked by two annotators with engineering backgrounds. When disagreements occurred, the final decision was made by experts who have won national or international first prizes in engineering modeling competitions, ensuring technical rigor and accuracy.

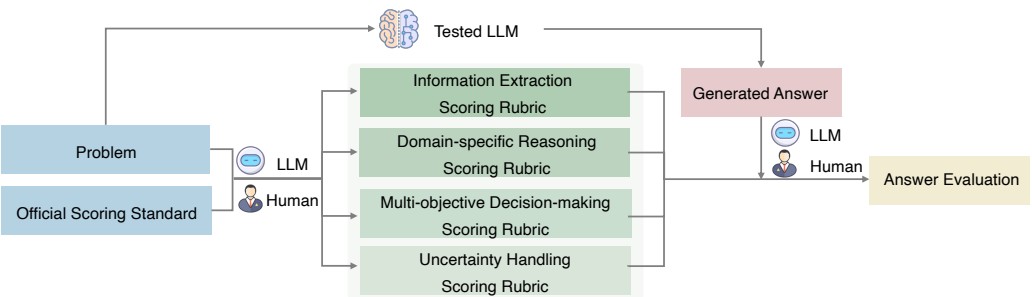

Figure 8: Workflow for generating the modified scoring rubrics. The official scoring standard and contest problem are first provided to a LLM to generate draft rubrics aligned with four core capabilities: Information Extraction, Domain-Specific Reasoning, Multi-Objective Decision-Making, and Uncertainty Handling. The resulting rubrics are then reviewed and refined by domain experts to ensure technical accuracy and alignment with modeling principles.

The prompt used for rubric generation is provided below:

```
1  """Assume you are an expert in problem design and grading, with deep
      familiarity with mathematical modeling. Please help me design an
      evaluation rubric for assessing large language models' engineering
      capabilities. Specifically, I will provide a problem and its scoring
      criteria, and you will tell me which of the following capabilities
      are assessed by this rubric: redundant_information_filtering_score,
      multi_objective_tradeoff_score, uncertainty_handling_score, and
      deep_knowledge_integration_score. In particular, please identify how
      each capability is assessed through specific aspects of the problem
      or rubric.
2
3  For each capability that is covered, provide a scoring rubric in the
      following format:
4
5  Problem [(Problem ID)]:
6  redundant_information_filtering_score: (1)(2)...
7  multi_objective_tradeoff_score: (1)(2)...
8  uncertainty_handling_score: (1)(2)...
9  deep_knowledge_integration_score: (1)(2)...
10
11 Notes: Each capability has a total possible score of 10 points. In other
      words, the total score for each listed capability should sum to 10
      points. Capabilities that are not covered in this problem receive 0
      points. The rubric should further specify, under each capability, the
       different score levels (e.g., 1 point, 2 points, 3 points, etc.) and
       the corresponding specific behaviors or response characteristics
      associated with each level.
12
13 Please read the problem and rubric carefully and provide a capability-
      based evaluation rubric for how this problem assesses the output of
      large language models."""
```

The prompt used to evaluate the generated answer against the rubric is as follows:

```
1  f"""
2      You are a professional modeling competition judge with extensive
       experience in evaluating mathematical and engineering models. Please
       conduct a rigorous evaluation of the following answer based on the
       provided criteria.
3
4      Answer to evaluate:
5      {answer}
6
7      Evaluation Criteria:
8      {score_criteria}
9
10     Please evaluate strictly according to the criteria and provide your
       assessment in the following JSON format:
11     {{
12         "score": <score between 0-10, can use decimal points for
       precision>,
13         "reason": "Detailed evaluation breakdown:\n
14                 1. [Specific criterion] - [sub-score] points: [
       justification]\n
15                 2. [Specific criterion] - [sub-score] points: [
       justification]\n
16                 3. [Specific criterion] - [sub-score] points: [
       justification]\n
17                 Final score: [total] points"
18     }}
19
20     Note:
21     - Break down your scoring into specific components
22     - Provide clear justification for each sub-score
23     - Be objective and consistent in your evaluation
24     - Consider both the technical accuracy and the methodology
25     """
```

### F.3 LEVEL 3 SCORING EXAMPLES

As results shown in section 4.2.3, the answers of LLMs to open-ended tasks show significant differences in four dimensions of information extraction, multi-objective decision making, uncertainty handling and domain-specific reasoning. Figure 7 preliminarily presents two scoring segments, 3 points and 8 points, for the evaluation of models' answers. To demonstrate the response performance of different segments more clearly and intuitively, we provide the following examples with more Level 3 scoring details:

1. **Full Mark (Avg. Score: 9.475):** The problem requires optimizing Hu sheep farm pen utilization under stochastic conditions (conception rates, gestation periods, litter sizes) while adhering to strict capacity constraints and cohabitation rules. The solution must minimize expected losses from idle pens (1 unit/day) or shortages (3 units/day) through dynamic scheduling and statistical validation.

   - **Information Extraction (10/10):**
     Exclusion of Deterministic Assumptions (5/5): Section 1 (System Overview) clarifies all critical parameters modeled as random variables (e.g., "$X_c \sim \text{Binomial}(N_m, 0.85)$: Number of successful conceptions; $G \sim U[147, 150]$: Gestation days; $L_s \sim \text{Poisson}(\lambda = 2.2)$: Liveborn lambs per ewe, with 3% mortality ($L_a = L_s \cdot 0.97$); $L_d \sim U[35, 45]$: Lactation days"). Section 3A (Scenario Generation) replaces fixed values with dynamic sampling (e.g., "For each scenario, sample: - Which ewes conceive (Bernoulli, 85%) - Their gestation ($G$) - Number of lambs ($L_s$), apply mortality - Lactation length ($L_d$)"). Section 6B (Robust Planning) makes flexible scheduling responsive to stochastic outcomes (e.g., "Adjust mating/rest period within allowed windows to shift animal flows.").
     Identification of Valid Uncertainty Parameters (5/5): Section 1 clarifies explicit distributions for all uncertainties (e.g., "$X_c \sim \text{Binomial}(N_m, 0.85)$... $G \sim U[147, 150]$...

$L_s \sim \text{Poisson}(2.2)$... $L_d \sim U[35, 45]$"). Section 3A ensures consistent application in scenario generation (e.g., "Sample conception (Bernoulli), gestation ($G$), litter size ($L_s$), lactation ($L_d$)."). Section 5 (Loss Function) offers loss calculation integrating stochastic inputs (e.g., "$\mathbb{E}_{scenario}[\sum_t[I_t + 3S_t]]$").

- **Multi-objective Decision making (9.2/10):**

  Minimized Expected Loss & Output Maximization (4.5/5): Section 5 (Loss Function) contains rigorous mathematical formulation balancing idle (1 unit) vs. shortage (3 unit) costs (e.g., "Objective: $\min \mathbb{E}_{scenario}[\sum_t[I_t + 3S_t]]$ $I_t =$ Idle pens, $S_t =$ Shortages"). Section 7B (Robust Planning) includes statistical validation of tradeoffs (e.g., "Monte Carlo over Scenarios: Simulate losses across all scenarios for each candidate policy.") Section 8 (Results Table) applies quantitative comparison of policies.

  Lactation Flexibility & Fattening Tradeoffs (4.7/5): Section 1 (System Overview) makes explicit dynamic linkage between lactation and fattening (e.g., "$L_d \sim U[35, 45]$: Lactation days $\to F_d = 210 + 2 \cdot (40 - L_d)$: Fattening days"). Section 6B (Robust Planning) considers operational use of flexibility to smooth demand (e.g., "Adjust rest periods to align cohorts, minimizing 'loner pens'."). Section 3A (Scenario Generation) has stochastic integration of tradeoff (e.g., "Sample lactation length ($L_d$), impact on fattening ($F_d$).").

- **Uncertainty Handling (9.2/10):**

  Stochastic Process Models (4/4): Section 1 (System Overview) specifies explicit distributions for all stochastic parameters (e.g., "$X_c \sim \text{Binomial}(N_m, 0.85)$, $G \sim U[147, 150]$, $L_s \sim \text{Poisson}(2.2)$, $L_d \sim U[35, 45]$"). Section 3A (Scenario Generation) implements full Monte Carlo (e.g., "Generate 1000 scenarios... sample conception (Bernoulli), gestation ($G$), litter size ($L_s$), lactation ($L_d$)."). Section 7B (Robust Planning) includes statistical validation of stochastic outcomes (e.g., "For each candidate policy, simulate losses across all scenarios.").

  Dynamic Adjustment Strategies (2.7/3): Section 1 (Fattening Calculation) establishes mechanistic linkage of lactation-fattening tradeoff (e.g., "$F_d = 210 + 2 \cdot (40 - L_d)$: Fattening days adjusted by lactation."). Section 6B (Robust Planning) makes adaptive scheduling but lacks two-way feedback (e.g., "Adjust rest periods to align cohorts... weekly rolling re-optimization.").

  Contingency Sets (2.5/3): Section 2 (Cohabitation Rules) contains hard-coded tolerance for uncertainty (e.g., "Group into largest feasible penfuls within 7-day windows."). Section 8 (Statistical Assessment) analyzes multi-scenario sensitivity (e.g., "Tabulate average loss, shortage probability, and max pen use.").

- **Domain-specific Reasoning (9.5/10):**

  Integration of Empirical Rules (4/4): Section 2 (Cohabitation Rules) adds hard-codes industry constraints into algorithms (e.g., "7-day tolerance window for nursing ewes, lambs, and resting ewes... Group into largest feasible penfuls (14 fattening lambs/pen, 6 nursing ewes/pen)."). Section 1 (System Overview) uses embeds empirical flexibility ranges as distributions (e.g., "$L_d \sim U[35, 45]$: Lactation days... $R \sim U[18, 22]$: Adjustable rest period.") Section 6B (Robust Planning) operationalizes flexible rest rules (e.g., "Extend rest periods to align cohorts if pens would otherwise idle.").

  Expected Loss Functions (3/3): Section 5 (Loss Function) has rigorous probabilistic loss aggregation (e.g., "$\min \mathbb{E}_{scenario}[\sum_t[I_t + 3S_t]]$ $I_t = \max(P_{avail} - P_{req}(t), 0)$, $S_t = \max(P_{req}(t) - P_{avail}, 0)$."). Section 8 (Results Table) quantifies loss distribution across scenarios. Section 3B (State Evolution) links stochastic occupancy to loss calculation (e.g., "For each day $t$: Compute $P_{req}(t)$ from sampled cohorts.").

  Stochastic Optimization Algorithms (2.5/3): Section 7B (Robust Planning) applies sample average approximation (SAA) method (e.g., "Monte Carlo simulation over 1000 scenarios to evaluate policies."). Section 6A (Rolling Horizon) uses heuristic dynamic programming (e.g., "Re-optimize mating batches weekly to maximize cohabitation.").

2. **5 points (Avg. Score: 5.375):** The problem involves modeling a team coordination exercise ("Unity Drum") where 8 members control a drum's tilt by pulling ropes to bounce a ball. Key tasks include: 1. Calculating the drum's tilt angle at t=0.1s based on force/timing inputs (Table 1), accounting for initial 11cm displacement. 2. Ensuring physics-based accuracy in torque, angular acceleration, and geometric relationships.

- **Information Extraction (7.5/10):**

  Error Source Analysis (5/6): Explicit Recognition: Timing errors-"Some members may apply force slightly before others" (Algorithm section); strength variation-"Members likely have different strengths" (Considerations). Partial Implementation: Timing logic in code (if timing$[i] \leq 0.1$) is noted but lacks vector-time coupling; force scaling (effective_force $= \frac{\text{force}(\text{member\_id}-1)}{10}$) is arbitrary.

  Physical Model Simplification (2.5/4): Justified Simplifications: "Ignores damping for short-duration calculation" (Considerations); Drum as uniform cylinder ($I = 0.5 \cdot \text{drum\_mass} \cdot r^2$). Over-Simplifications: Fixed torque angle ($\sin\left(\frac{\pi}{2}\right)$) ignores vector geometry; rope tautness assumption ("If the drum tilts too far, ropes could slack") not modeled.

- **Multi-objective Decision making (6.5/10):**

  Tilt Angle and Force Relationship (4.5/6): Physics Foundation: Correctly derives torque ($\tau = r \cdot F \cdot \sin(\theta)$), inertia ($I = 0.5 \cdot m \cdot r^2$), and angular kinematics ($\theta = \theta_0 + \frac{1}{2}\alpha t^2$); maps rope geometry (angle_radians $= (\text{member\_id} - 1) \cdot \left(\frac{2\pi}{8}\right)$). Implementation Gaps: Timing logic (if timing$[i] \leq 0.1$) is crude; forces are binary (on/off) rather than time-interpolated; no optimization for tilt minimization (e.g., predictive control or force balancing).

  Computational Efficiency (2/4): Basic Looping-iterates over 8 members with O(1) operations per member (e.g., torque $=$ drum_radius $\cdot$ force $\cdot \sin\left(\frac{\pi}{2}\right)$). No Advanced Techniques-lacks vectorization, memoization, or scalability for larger teams.

- **Uncertainty Handling (2/10):**

  Error Propagation Analysis (2/4): Acknowledgment Only: Mentions "members likely have different strengths and reaction times" (Considerations); suggests "extended to simulate more realistic distributions" but provides no math or implementation. No Quantification: Lacks sensitivity analysis or error bounds on tilt angle.

  Numerical Simulation Estimation (0/4): No Monte Carlo: Code calculates tilt for fixed inputs only (force_data); no randomization of force/timing or statistical output (mean/variance).

  Methodological Clarity (N/A): Physics steps are clear but irrelevant to uncertainty scoring.

- **Domain-specific Reasoning(5.5/10):**

  3D Mechanics Modeling (2.5/6): 2D Limitation: Explicitly states "our coordinate system will be planar (X and Y only)" (Key Equations); torque calculation ($\tau = r \cdot F \cdot \sin(\theta)$) ignores out-of-plane forces. Partial Physics: Correctly models drum as cylinder ($I = 0.5 \cdot m \cdot r^2$) but lacks 3D rotation dynamics.

  Model-Based Optimization Strategy (3/4): Suggestions Without Implementation: Proposes "damping term proportional to angular velocity" (Considerations); mentions "member variation" but no adaptive control (e.g., PID for tilt correction).

3. **1 point (Avg. Score: 1.25):** The problem involves coordinating multiple meteorological units (each with 1 primary and 2 secondary stations) to ensure reliable hourly weather data collection and full data sharing under strict communication constraints. Key challenges include managing transmission reliability (80% for secondaries, 100% for primaries), message capacity limits, and achieving 97% success probability within 8 minutes for primary data exchange. The goal is to determine the maximum number of units (Nmax), design transmission schemes, and compute performance metrics.

   - **Information Extraction (2/10):** High-Probability Constraint Processing (0/5): Failure to Address Probabilistic Guarantee: The answer calculates secondary transmission success as "expected number of reports received... is $4 \times 0.8 = 3.2$" (Step 4) but never models retransmissions or redundancy to achieve 97% success. The assumption of direct success ignores the problem's explicit probability requirement. Missing Critical Logic: No discussion of how to compensate for the 20% failure rate (e.g., retrying failed transmissions, acknowledgments, or error correction).

     Time Window Isolation (2/5): Interleaved Logs Without Justification: The primary and secondary transmission logs (Tables 1 2) are interleaved in the solution ("Round 1: Primary 1→2; Round 1: Secondary 1→1a"), but no protocol ensures collision

avoidance (e.g., TDMA, priority scheduling). Unverified Simultaneity Assumption: The answer states "Simultaneous reception allowed during transmission" (Step 1) but doesn't prove this suffices for concurrent primary/secondary transmissions under the 8-minute constraint.

- **Multi-objective Decision making (2/10):**

   3D Parameter Optimization (0/6): Single-Parameter Focus: The answer only optimizes for **N_max** ("$N(N-1)/28 \to N_{\max} = 4$", Step 2) but ignores joint optimization of capability (no analysis of 158-character message limits or segment splitting efficiency), reliability (no adjustment for secondary station 80% success rate such as no retransmission strategy) and time (assumes 8 minutes suffice without validating secondary transmission overhead). Missed Pareto Frontier: Fails to explore tradeoffs (e.g., "Could N=5 work if secondary transmissions are reduced?").

   Resource Allocation Strategy (2/4): Equal Bandwidth Only: Primary stations follow a round-robin schedule ("1→2, 1→3, 1→4, 2→3, ...", Table 1), and secondaries transmit uniformly ("1→1a, 1→1b, 2→2a, ...", Table 2). No Prioritization: Critical objectives (e.g., ensuring 97% success) aren't prioritized in scheduling.

- **Uncertainty Handling (0/10):**

   High-Order Probability Events (0/6): No Threshold Calculation: The answer states secondary stations have an "80% transmission/reception success rate" (Step 1) but never computes the probability of achieving 97% success (e.g., via binomial distribution for multiple retries). Misleading Metric: The "mean secondary reports received per primary station (3.2)" (Step 4) is irrelevant to the cumulative success probability requirement.

   Asymmetric Loss (0/4): No Cost Analysis: The solution ignores idle time cost (unused transmission slots due to failures) and rental loss (penalties for delayed data delivery implied by "critical rescue operations").

- **Domain-specific Reasoning (1/10):**

   Mixed-Integer Programming (0/5): No Optimization Model: The answer derives $N_{\max} = 4$ via a simple inequality ("$\frac{N(N-1)}{2} \leq 8$", Step 2) but lacks an objective function (e.g., "maximize N while meeting time/reliability constraints"), and omits integer constraints (N must be discrete) or linear relaxation techniques. Ad-Hoc Calculation: No use of MINLP (Mixed-Integer Nonlinear Programming) to jointly optimize N, transmission scheduling, and reliability.

   Fault-Tolerant Protocol Design (1/5): Basic Segmentation: Mentions "reports can split into two 50-character segments" (Step 1) but no dual verification (never states if segments are sent redundantly to different primaries) and no formal protocol (assumes secondary stations report to all primaries without fault recovery like checksums, ACKs).

## G  ADDITIONAL ANALYSIS

### G.1  LEVEL 1 ANALYSIS

**Minor perturbations cause performance drops, revealing shallow generalization.**  Figure 9 (left) presents model accuracy on Level 1 tasks across four input variants: Original, Perturbed, Knowledge-enhanced, and Math Abstraction. When problems are perturbed through minor changes in wording or numerical values, average model accuracy drops from 82.9% to 81.5%. Notably, Llama 3.3 and Qwen2.5-72B decline by 6.6% and 5.1%, respectively. This indicates that some models exhibit limited robustness and often rely on memorized phrasing or surface patterns rather than generalizable reasoning.

**Explicit knowledge prompts mitigate reasoning failures in weaker models.**  When explicit domain knowledge—such as formulas, constants, or unit conversions—is added to the input, accuracy improves to 85.5% on average. Weaker models benefit the most: GPT-4.1 Mini gains 6.2% and Mixtral-8x7B improves by 9.3%. This pattern suggests that many errors are not caused by a complete lack of knowledge, but rather by the inability to retrieve and apply relevant concepts without targeted prompting. Explicitly embedding domain knowledge thus serves as an effective intervention for enhancing reasoning activation.

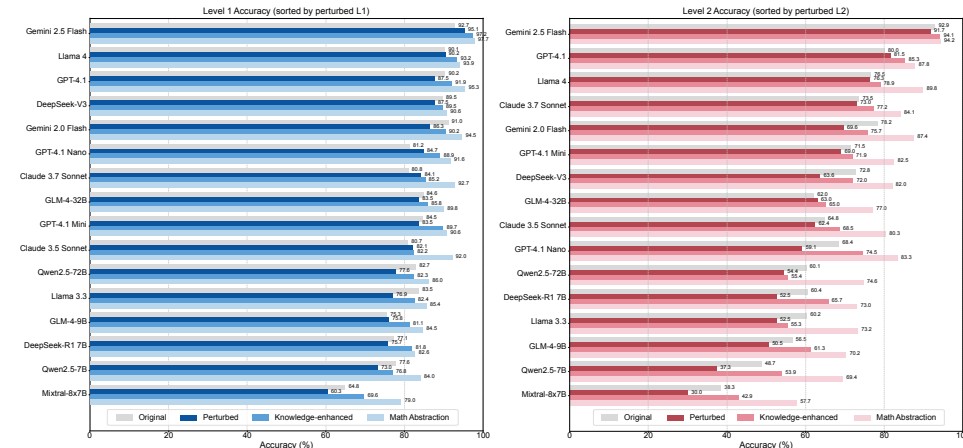

Figure 9: Accuracy of LLMs on Level 1 (left) and Level 2 (right) tasks across four variants: Original, Perturbed, Knowledge-enhanced, and Math Abstraction. Drops in the Perturbed version indicate sensitivity to input changes, while gains in the latter two show that current LLMs require external knowledge or reformulation to improve accuracy—highlighting their lack of these abilities.

**Removing contextual language highlights semantic limitations.** Performance further increases to 89.4% when problems are rewritten into abstract mathematical form, removing all contextual language. For example, Qwen2.5-7B and Mixtral-8x7B improve by 10.9% and 18.8%, respectively. This reveals that most Level 1 failures are not due to weak computational ability, but rather arise during semantic interpretation and variable binding. Once language ambiguity is removed, models can more reliably execute the required calculations, underscoring a gap between symbolic proficiency and contextual understanding.

## G.2 LEVEL 2 ANALYSIS

Level 2 tasks emphasize multi-step reasoning under structured constraints, making them more sensitive to input variability. As shown in Figure 9 (right), the average model accuracy declines from 66.6% on the Original version to 61.6% on the Perturbed variant. This 5.0% drop indicates that even minor changes to semantic phrasing or numerical values can significantly disrupt reasoning chains. For instance, GPT-4.1 Nano drops by 9.3% and Qwen2.5-7B by 11.4%, revealing their limited robustness when facing contextual and structural perturbations in problem inputs.

Incorporating explicit domain knowledge helps reduce ambiguity and recover performance. With knowledge-enhanced inputs, the average accuracy rises to 68.6%, a 7.0% improvement over the perturbed baseline. Larger gains are observed for models such as GPT-4.1 Nano (+15.4%) and Qwen2.5-7B (+16.6%), suggesting that knowledge prompts assist in constraint interpretation and formula selection. However, some models such as DeepSeek-V3 show minimal improvement, implying that knowledge access alone may not compensate for limitations in multi-step reasoning capabilities.

Symbolic abstraction of Level 2 tasks into pure mathematical form results in the largest performance gains. The average accuracy increases to 79.2%, with many models gaining over 15%. This trend is especially prominent for weaker models like Qwen2.5-7B (from 37.3% to 69.4%) and Mixtral-8x7B (from 30.0% to 57.7%). These improvements confirm that many model failures stem not from computational weakness, but from difficulties parsing, organizing, and executing the reasoning steps embedded in natural language problem statements. This underscores the importance of assessing upstream cognitive processes that precede symbolic computation—dimensions often underexamined in traditional mathematical benchmarks.

## G.3 LEVEL 3 ANALYSIS

Figure 10 presents the performance of various models across four key capabilities: Redundant Information, Multi-Objective Decision, Domain Knowledge, and Uncertainty Handling. The results are further separated into *original* and *rewritten* problem formulations. Overall, human experts substantially outperform all models across all dimensions, with average scores of 8.552 (original) and

| | Redundant Info (Original) | Redundant Info (Rewritten) | Multiobjective (Original) | Multiobjective (Rewritten) | Knowledge (Original) | Knowledge (Rewritten) | Uncertainty (Original) | Uncertainty (Rewritten) | Average (Original) | Average (Rewritten) |
|---|---|---|---|---|---|---|---|---|---|---|
| Human Expert | 9.134 | 9.064 | 8.295 | 8.370 | 8.701 | 8.590 | 8.782 | 8.921 | 8.552 | 8.736 |
| GPT-4.1 | 8.334 | 8.365 | 7.419 | 7.290 | 6.598 | 6.437 | 6.081 | 5.916 | 7.108 | 7.002 |
| Claude 3.7 Sonnet | 8.349 | 8.185 | 6.096 | 6.716 | 6.229 | 5.529 | 5.951 | 5.351 | 6.656 | 6.445 |
| GPT-4.1 Mini | 7.793 | 7.528 | 6.880 | 6.203 | 6.179 | 5.984 | 6.322 | 5.798 | 6.793 | 6.378 |
| DeepSeek-V3 | 7.561 | 7.570 | 6.375 | 6.521 | 5.850 | 5.862 | 5.369 | 5.361 | 6.289 | 6.329 |
| Gemini 2.5 Flash | 8.320 | 7.652 | 6.434 | 6.263 | 4.746 | 5.533 | 5.287 | 5.383 | 6.197 | 6.208 |
| Gemini 2.0 Flash | 6.878 | 7.015 | 6.188 | 6.067 | 6.145 | 5.892 | 5.370 | 4.990 | 6.145 | 5.991 |
| GPT-4.1 Nano | 7.118 | 6.536 | 6.080 | 6.154 | 5.236 | 5.206 | 5.438 | 5.058 | 5.968 | 5.738 |
| GLM-4-32B | 6.937 | 6.583 | 5.902 | 5.579 | 4.512 | 5.715 | 5.110 | 4.733 | 5.615 | 5.653 |
| GLM-4-9B | 5.796 | 5.959 | 4.813 | 5.399 | 4.397 | 4.734 | 4.328 | 4.586 | 4.833 | 5.169 |
| Claude 3.5 Sonnet | 6.444 | 6.271 | 5.774 | 5.190 | 4.173 | 4.531 | 4.520 | 4.204 | 5.228 | 5.049 |
| Llama 3.3 | 5.840 | 5.995 | 4.965 | 5.072 | 4.385 | 4.673 | 4.157 | 4.290 | 4.837 | 5.008 |
| Qwen2.5-72B | 6.075 | 5.696 | 4.896 | 4.819 | 4.479 | 4.313 | 4.289 | 4.061 | 4.935 | 4.722 |
| Qwen2.5-7B | 5.094 | 5.541 | 4.688 | 4.731 | 3.934 | 4.218 | 3.641 | 3.988 | 4.339 | 4.619 |
| Llama 4 | 5.562 | 5.753 | 3.076 | 3.311 | 3.519 | 3.081 | 2.912 | 3.299 | 3.767 | 3.861 |
| DeepSeek-R1 7B | 5.051 | 5.168 | 4.015 | 3.541 | 3.254 | 3.621 | 3.853 | 2.910 | 4.043 | 3.810 |
| Mixtral-8x7B | 3.678 | 4.443 | 3.504 | 3.319 | 3.318 | 3.248 | 2.311 | 2.893 | 3.203 | 3.476 |

Figure 10: Level 3 Model Evaluation. The figure presents average model performance on Level 3 tasks across four capability dimensions—information extraction, domain-specific reasoning, multi-objective decision-making, and uncertainty handling—under both original and rewritten problem formulations.

8.736 (rewritten). In contrast, LLMs demonstrate significantly lower scores, revealing a persistent gap between current LLMs' capabilities and human-level reasoning. The average model scores before and after rewriting are 5.663 and 5.617, respectively—a marginal difference of only 0.81%. This indicates that most models possess a reasonable degree of generalization, and the benchmark shows no signs of data contamination across reformulated prompts, preserving task consistency.

Based on the overall average scores, we categorize model performance into three tiers:

**Tier 1 (Average Score > 6.5)** This tier includes GPT-4.1, Claude 3.7 Sonnet, and GPT-4.1 Mini. These models demonstrate strong performance across all four evaluated capabilities. In particular, their scores in Information Extraction and Multi-Objective Decision often exceed 7, approaching human expert levels. Their performance in Domain Knowledge and Uncertainty Handling also remains consistently above 6, indicating robust reasoning capabilities and broad task adaptability.

**Tier 2 (Average Score ≈ 5.7–6.5)** This tier consists of DeepSeek-V3, Gemini 2.5 Flash, Gemini 2.0 Flash, GPT-4.1 Nano, and GLM-4-32B. These models achieve reasonable performance in Information Extraction and Multi-Objective Decision, but exhibit noticeable weaknesses in Domain Knowledge and Uncertainty Handling, where scores commonly fall below 6. Some models approach the 5-point threshold in these dimensions, reflecting limitations in complex reasoning and knowledge integration.

**Tier 3 (Average Score < 5.7)** This tier includes GLM-4-9B, Claude 3.5 Sonnet, Llama 3.3, Qwen2.5-72B, Qwen2.5-7B, Llama4, DeepSeek-R1 7B, and Mixtral-8x7B. These models consistently underperform across all four capabilities, typically scoring between 3 and 5. Their weakest areas are Domain Knowledge and Uncertainty Handling, where some models fall below 4. These results indicate substantial deficiencies in background reasoning and generalization to ambiguous or underspecified tasks.

## G.4 SUBFIELD PERFORMANCE ANALYSIS

**Model performance varies substantially across engineering subfields.** Chemical and biological engineering demonstrates the strongest robustness, with large models maintaining accuracies above 85%, while structural and physical engineering achieves 70–80% and systems and control engineering performs the worst, with large models dropping to 60–70% and small models often below 40%. These results suggest that robustness to contextual perturbations is closely tied to the task characteristics:

chemical and biological problems rely more on formulaic knowledge and are less sensitive to input variations, whereas systems and control problems involve more complex reasoning chains and are more vulnerable to perturbations.

**Problem variants reveal subfield-specific differences in knowledge use, reasoning, and robustness, showing that these abilities differ significantly between engineering domains.** The knowledge-enhanced variant substantially improves performance in chemical and biological engineering, moderately benefits structural and physical engineering, and shows limited gains in systems and control engineering, suggesting the latter's inability to effectively leverage explicit knowledge. Similarly, the math abstraction variant, which isolates mathematical reasoning by removing context, favors chemical and biological engineering, followed by structural and physical engineering, while systems and control engineering remains the weakest. These patterns indicate that the ability to utilize injected knowledge and maintain mathematical reasoning varies considerably across subfields.

**The robustness and capability differences across subfields become even more evident under higher task complexity in Level 2.** Compared to Level 1, Level 2 shows larger performance drops under perturbed inputs, highlighting more severe robustness issues. The positive effects of knowledge-enhanced and math abstraction variants remain concentrated in chemical and biological engineering, with only marginal improvements in structural and physical engineering and negligible gains in systems and control engineering. This indicates that in more complex reasoning and contextual integration tasks, current large language models struggle even more to handle input perturbations, exploit external knowledge effectively, and maintain consistent reasoning, further widening the capability gap across subfields.

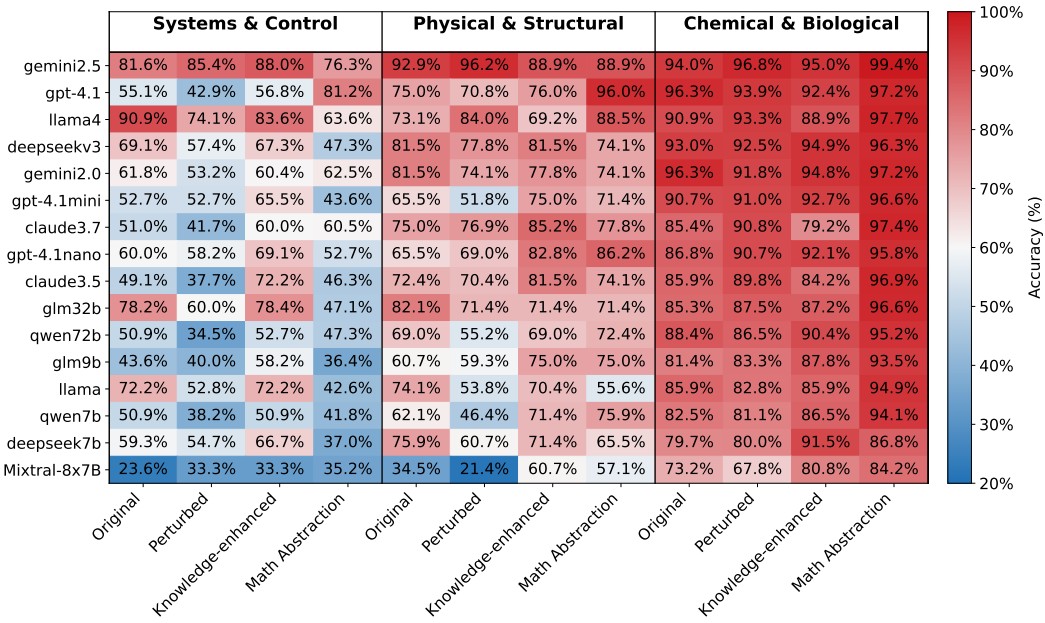

Figure 11: Accuracy across engineering subfields and problem variants in Level 1.

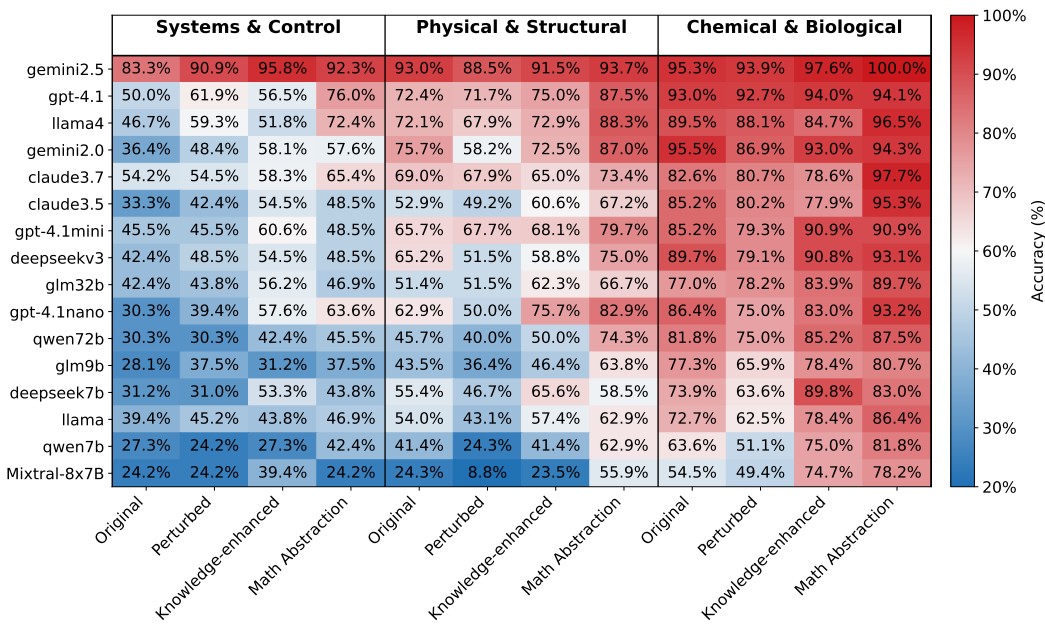

Figure 12: Accuracy across engineering subfields and problem variants in Level 1.

