# OpenReview forum: "EngiBench: A Benchmark for Evaluating Large Language Models on Engineering Problem Solving"
_ICLR.cc/2026/Conference — ICLR 2026 Conference Withdrawn Submission_

### Official Review · Reviewer_Zo8M · 2025-10-26

**Soundness:** 3
**Presentation:** 3
**Contribution:** 3
**Rating:** 6
**Confidence:** 3

**Summary:**

This paper introduces EngiBench, a benchmark for evaluating LLMs on solving real-world engineering problems. The benchmark spans three difficulty levels 1) foundational knowledge retrieval, 2) contextual reasoning, and 3) open-ended modeling. Together, they cover multiple engineering subfields including systems & control, physical & structural, and chemical/biological. Each problem is further rewritten into three controlled variants to diagnose robustness, knowledge use, and mathematical reasoning respectively.

The experiments contain 16 latest models, including GPT-4.1, Claude 3.7, Gemini 2.5, DeepSeek-V3, and open-source models like Qwen2.5 and Llama 4. Benchmarking results reveal that LLMs perform well on simple, structured problems but degrade significantly as tasks become open-ended. Perturbations cause notable accuracy drops, highlighting weak generalization, and even SOTA models fall short of human experts on Level 3 tasks. Overall, EngiBench demonstrates that LLMs lack the reliable high-level reasoning required for real-world engineering problems.

**Strengths:**

- The problem framing is practical and novel. It moves beyond symbolic reasoning toward real-world engineering problems, which was previously underexplored.

- The design with three difficulty levels enables granular capability analysis across different dimensions.

- The comparison is comprehensive, including SOTA models and human expert baselines.

- Codebase is submitted for reproducibility.

**Weaknesses:**

Overall, I am positive about the paper's contribution. Some minor weaknesses include:

- Scoring rubrics are constructed using LLMs to transform raw scoring descriptions into a capability-oriented rubric, this step may potentially introduce LLM bias and may not generalize across all engineering domains.

- Open-ended evaluation and human scoring might limit scalability when this benchmark is further developed, for example, include multimodal information.

**Questions:**

- How sensitive are Level 3 scores to the rubric granularity?

- Did you observe specific subfields, for example, physical problems, where LLMs perform better or worse?

- What would you expect about integrating multimodal information, like figures or diagrams? Will that change the conclusions about reasoning depth?

---

> ### Author Response · Authors · 2025-12-03
> **Response to Reviewer Zo8M**
>
> Dear Reviewer Zo8M,
>
> Thank you for your positive evaluation and constructive comments. We provide our detailed responses below.
>
> ## Weaknesses
>
> >**W1: "Scoring rubrics are constructed using LLMs to transform raw scoring descriptions into a capability-oriented rubric, this step may potentially introduce LLM bias and may not generalize across all engineering domains."**
>
> A1: Thank you for the comment. We clarify that our Level 3 rubric is not generated by LLMs. All **scoring criteria come directly from the official grading standards** of each problem, and the rubric is **structured and finalized by domain experts**. **LLMs play only an auxiliary role** in organizing information and do not determine the scoring rules, so LLM bias is not introduced. Since the tasks span multiple engineering subfields and all scoring points originate from their respective official standards, the rubric also maintains good cross-domain generality.
>
>
> >**W2: "Open-ended evaluation and human scoring might limit scalability when this benchmark is further developed, for example, include multimodal information."**
>
> A2: Thank you for the suggestion. Open-ended evaluation and human scoring do require manual effort, but they are also a key strength of our benchmark because such tasks capture important engineering capabilities that need expert-grounded evaluation standards. In this work, human scoring was used mainly for quality assurance. As shown in Section 3.3, our rubric already supports fully automatic evaluation, and the automatic scores align closely with human ratings, meaning future users can rely on automated evaluation when resources are limited. To ensure the quality of this version, over 20 domain experts participated in the annotation. We will continue to expand the benchmark.
>
>
> ## Questions
>
> >**Q1: "How sensitive are Level 3 scores to the rubric granularity?"**
>
> A1: Thanks for the question. We acknowledge that changes in rubric granularity may introduce minor variations, but such effects are very small. This reflects a common and reasonable trade-off in engineering evaluation, where a slight increase in granularity leads to much higher robustness, consistency, and cross-rater stability.
>
> Our Level 3 scoring is grounded in the official scoring criteria of each problem. These official scoring points define the basis for correctness and point assignment. Domain experts then organized and clarified these points without altering their meaning, ensuring that they can be applied consistently and reproducibly.
>
> In this setup, rubric refinement does not change the underlying scoring basis or introduce new criteria. It mainly reduces ambiguity and limits subjective variation among evaluators. As a result, although the level of detail in presentation may vary, the actual scoring logic and outcomes remain largely determined by the original official criteria, and any impact from granularity is minimal.
>
> This design improves inter-rater agreement and allows LLM-based scoring to align closely with human judgments, ensuring stable and reliable Level 3 evaluation.
>
>
> >**Q2: "Did you observe specific subfields, for example, physical problems, where LLMs perform better or worse?"**
>
> A2: Thank you for the question. Our results indicate **clear performance differences across engineering subfields**. As analyzed in **Appendix G.4 (Figures 11–12)**, chemical and biological engineering tasks show the highest robustness and benefit more from knowledge-enhanced variants, whereas structural/physical and systems & control problems are more sensitive to perturbations and struggle to leverage explicit engineering knowledge. This suggests that the ability to utilize domain knowledge and maintain stable reasoning varies significantly across engineering subfields.
>
>
> >**Q3: "What would you expect about integrating multimodal information, like figures or diagrams? Will that change the conclusions about reasoning depth?"**
>
> A3: Thank you for raising this important question. We believe that incorporating multimodal information such as figures and diagrams would enrich the problem representation and, in some cases, increase the complexity and realism of engineering tasks. This could allow models to leverage additional structural and spatial information during reasoning.
>
> However, we do not expect the overall conclusions on reasoning depth to change fundamentally. The current version of EngiBench **does not ignore multimodality**; for problems that originally contain graphical elements, we have already extracted and **converted the essential visual information into equivalent textual or structured representations to ensure fair comparison across models**. As a result, multimodal inputs may further enrich task descriptions, but are unlikely to alter the core findings regarding the depth of engineering reasoning. In future versions, we plan to include full multimodal engineering tasks to better reflect real-world scenarios.

---

### Official Review · Reviewer_1cnm · 2025-10-29

**Soundness:** 3
**Presentation:** 3
**Contribution:** 3
**Rating:** 4
**Confidence:** 4

**Summary:**

This paper introduces EngiBench, a benchmark designed to evaluate LLMs’ ability to solve real-world engineering problems, which require not only mathematical computation but also contextual understanding, multi-objective trade-offs, and uncertainty handling.
The benchmark covers three hierarchical difficulty levels (from foundational engineering knowledge to open-ended modeling tasks) and spans multiple engineering domains. The authors further construct perturbed, knowledge-enhanced, and math-abstraction variants for fine-grained diagnosis of model weaknesses such as robustness or missing domain knowledge. They evaluate a wide range of LLMs and observe a strong performance drop when moving to more realistic and open-ended tasks. Overall, the benchmark exposes key limitations of current LLMs in engineering problem-solving.

**Strengths:**

1.The paper addresses an important but underexplored question: Do LLMs actually have the ability to solve real engineering problems, rather than just math? This goes beyond existing benchmarks focused solely on abstract mathematics.

2. Incorporation of open-ended engineering questions: Level-3 tasks reflect real-world scenarios with incomplete constraints and trade-offs, making the benchmark more practical and challenging.

3. The benchmark includes multiple problem variations, enabling detailed analysis of robustness, mathematical abstraction, and knowledge gaps.

4. Diverse model evaluation: A strong suite of models—across sizes and accessibility—provides credible comparison and demonstrates wide applicability.

**Weaknesses:**

1.Lack of evaluation with math-specialized or science-oriented models,
It would be insightful to test models like Math-specialized LLMs or scientific reasoning model to assess whether their strong math/science capabilities transfer to engineering problems.

2.Missing training-side investigations,
Since Level-3 is extremely challenging, the authors could explore how fine-tuning or data augmentation improves performance, offering insights into how LLMs might better acquire engineering reasoning.

3. Presentation issues:Some visual elements reduce readability:
-Reference hyperlinks in bright green are visually distracting
-Figure 4 font size is too small to read clearly
Better formatting would improve the overall user experience.

**Questions:**

Mention in aboved issues.

---

> ### Author Response · Authors · 2025-11-28
> **Response**
>
> Dear Reviewer 1cnm,
>
> Thank you for your valuable comments. We provide our detailed responses below.
>
> >**W1: "Lack of evaluation with math-specialized or science-oriented models, It would be insightful to test models like Math-specialized LLMs or scientific reasoning model to assess whether their strong math/science capabilities transfer to engineering problems."**
>
> A1: We appreciate the reviewer’s suggestion. Due to computational constraints, we could not include a broad range of math-specialized models in this version. To preliminarily examine whether strong math/science reasoning transfers to engineering tasks, we added a reasoning model, Gemini 2.5 Pro, to our evaluation.
>
> As shown in Tables 1–3:
>
> - Level 1: Gemini 2.5 Pro performs similarly to Gemini 2.5 Flash, suggesting that basic structured tasks are close to saturation.
> - Level 2: Gemini 2.5 Pro achieves moderate gains, indicating that advanced reasoning helps contextual inference.
> - Level 3: Gemini 2.5 Pro shows a substantial improvement, performing comparably to GPT-4.1 and outperforming it on some engineering capabilities. However, it remains significantly below human experts.
>
> These results imply that while strong math/science reasoning contributes to improved engineering performance, it alone is insufficient for open-ended modeling and feasibility-constrained reasoning, reinforcing the necessity and challenge of our Level 3 tasks.
>
> **Table 1.** Performance of Gemini 2.5 Pro and selected models on Level 1 tasks.
>
> |  | Original | Perturbed | Knowledge-enhanced | Math Abstraction |
> | --- | --- | --- | --- | --- |
> | Gemini 2.5 Pro | 94.6 | 95.5 | 97.3 | 97.5 |
> | Gemini 2.5 Flash | 92.7 | 95.1 | 97.2 | 97.7 |
>
> **Table 2.** Performance of Gemini 2.5 Pro and selected models on Level 2 tasks.
>
> |  | Original | Perturbed | Knowledge-enhanced | Math Abstraction |
> | --- | --- | --- | --- | --- |
> | Gemini 2.5 Pro | 93.2 | 93.1 | 94.5 | 94.7 |
> | Gemini 2.5 Flash | 92.9 | 91.7 | 94.1 | 94.2 |
>
> **Table 3.** Performance of Gemini 2.5 Pro and selected models on Level 3 tasks.
>
> | Human Expert | Redundant Info (Original) | Redundant Info (Perturbed) | Multiobjective (Original) | Multiobjective (Perturbed) | Knowledge (Original) | Knowledge (Perturbed) | Uncertainty (Original) | Uncertainty (Perturbed) | Average (Original) | Average (Perturbed) |
> | --- | --- | --- | --- | --- | --- | --- | --- | --- | --- | --- |
> | Human | 9.134 | 9.064 | 8.295 | 8.370 | 8.701 | 8.590 | 8.782 | 8.921 | 8.552 | 8.579 |
> | GPT-4.1 | 8.334 | 8.365 | 7.419 | 7.290 | 6.598 | 6.437 | 6.081 | 5.916 | 7.108 | 7.002 |
> | Gemini 2.5 Pro | 8.868 | 8.179 | 7.403 | 7.331 | 6.176 | 6.257 | 6.260 | 6.071 | 7.177 | 6.960 |
> | Gemini 2.5 Flash | 8.320 | 7.652 | 6.434 | 6.263 | 4.746 | 5.533 | 5.287 | 5.383 | 6.197 | 6.208 |
>
>
> >**W2: "Missing training-side investigations, Since Level-3 is extremely challenging, the authors could explore how fine-tuning or data augmentation improves performance, offering insights into how LLMs might better acquire engineering reasoning."**
>
> A2: We appreciate the valuable suggestion. The primary focus of this work is to construct a diagnostic benchmark capable of evaluating comprehensive engineering problem-solving capabilities. While training-side explorations such as fine-tuning and data augmentation are crucial for addressing Level 3 challenges, conducting them without a rigorous evaluation framework risks confounding genuine reasoning improvements with template memorization. Therefore, we believe establishing this credible evaluation standard is the necessary prerequisite, and we plan to investigate these training strategies in our future work.
>
> >**W3: "Presentation issues:Some visual elements reduce readability: -Reference hyperlinks in bright green are visually distracting -Figure 4 font size is too small to read clearly Better formatting would improve the overall user experience."**
>
> A3: We thank the reviewer for the helpful suggestion. In the revised version, we will refine the visual presentation by (1) adjusting reference link colors to reduce distraction, and (2) enlarging the font size and improving readability in Figure 4. We appreciate the feedback and will ensure a clearer user experience.

---

### Official Review · Reviewer_2aWb · 2025-10-31

**Soundness:** 2
**Presentation:** 2
**Contribution:** 2
**Rating:** 2
**Confidence:** 4

**Summary:**

This paper introduces EngiBench, a new benchmark designed to evaluate large language models (LLMs) on their ability to solve real-world engineering problems. The authors argue that existing benchmarks focus too narrowly on well-posed mathematical problems and fail to capture the complexities of real-world engineering, such as uncertainty, context, and open-ended scenarios.
EngiBench is a hierarchical benchmark with three levels of increasing difficulty:
Level 1: Foundational knowledge retrieval, which involves applying basic formulas in a single step.
Level 2: Multi-step contextual reasoning under defined constraints.
Level 3: Open-ended modeling, requiring information extraction, domain-specific reasoning, multi-objective decision-making, and uncertainty handling.
To conduct a fine-grained analysis of model performance, the problems in Levels 1 and 2 are rewritten into three controlled variants: perturbed, knowledge-enhanced, and math abstraction. This allows the evaluation of a model's robustness, domain-specific knowledge, and mathematical reasoning abilities separately.
The paper's experiments show a clear performance gap across these difficulty levels, with models struggling more as tasks become harder. Results reveal that even state-of-the-art LLMs perform significantly worse than human experts on the high-level, open-ended tasks in Level 3, indicating that they currently lack the necessary high-level reasoning for real-world engineering problems.

**Strengths:**

- The idea of creating a benchmark focused on real-world engineering problems is good and rooted in application of LLMs in practical settings. The creation of a hierarchical benchmark, EngiBench, with three distinct difficulty levels from foundational knowledge retrieval to open-ended modeling is also good.
- The methodology of systematically generating controlled variants for each problem is interesting and acts as a diagnostic tool to pinpoint specific weaknesses in models.
- For the complex, open-ended tasks in Level 3, the use of expert-designed scoring rubrics and annotation by PhD students and professionals provides a reliable and rigorous evaluation framework.
- Evaluation across diverse models and task levels is thorough and the controlled variants help in analysing specific gaps in models and bring out insights.

**Weaknesses:**

- Level 1 and 2 problems are borrowed from existing benchmark datasets and follow a simple binary scoring evaluation. The non-trivial contribution is mainly the level 3 problems. This limits the overall contribution of the work. This does not agree with the authors' initial motivation that highlighting the gaps in the existing benchmarks.
- The paper acknowledges that many authentic engineering problems are inherently multimodal, relying on visual elements such as diagrams, schematics, and structured tables. By restricting the benchmark to text-only formats to ensure consistency, the evaluation overlooks a crucial aspect of engineering reasoning that involves interpreting and integrating information from both visual and textual sources.
- It is unclear if the benchmark covers long-context problems. Real-world engineering tasks often involve navigating extensive documentation and large datasets.
- Some of the insights around models performing better on knowledge-enhanced variants or low reasoning capabilities of smaller-scale models are fairly acknowledged already.

**Questions:**

- On page 7, you say that - "we introduce perturbed variants that modify surface details but keep the core structure unchanged" -- isn't this  already done as part of the dataset creation?
- There's some repetition in the writing. For instance, "We define this as engineering problem-solving ability, comprising four key dimensions: information extraction, domain-specific reasoning, multi-objective decision-making, and uncertainty handling" - this is introduced multiple times.

**Details Of Ethics Concerns:**

The work leverages human annotators for benchmark creation. The ethical considerations around this are not discussed in the paper.

---

> ### Author Response · Authors · 2025-11-28
> **Response (1/3)**
>
> Dear Reviewer 2aWb,
>
> Thank you for the insightful comments. We provide our detailed responses below.
>
> ## Weaknesses
>
> >**W1: "Level 1 and 2 problems are borrowed from existing benchmark datasets and follow a simple binary scoring evaluation. The non-trivial contribution is mainly the level 3 problems. This limits the overall contribution of the work. This does not agree with the authors' initial motivation that highlighting the gaps in the existing benchmarks."**
>
> A1: We appreciate the reviewer’s recognition of Level 3. We clarify that Level 1–2 are not simply borrowed from existing benchmark datasets; they are essential components designed to diagnose engineering‐specific reasoning gaps that existing benchmarks cannot evaluate.
>
> Level 1 tests cross-disciplinary engineering knowledge invocation under minimal context, while Level 2 adds realistic engineering constraints (e.g., physical limits, unit consistency, variable coupling, conservation), requiring interpretable multi-step reasoning beyond formula recall. Level 3 focuses on open-ended modeling. Thus, **the three levels capture complementary**, not incremental, dimensions of engineering capability.
>
> Moreover, Level 1–2 are not merely derived from existing datasets. As detailed in Appendix D.1, they were constructed via:
> 1. collection of 100K+ raw problems,
> 2. domain-based filtering into appropriate subfields,
> 3. semantic/numerical rewriting and three diagnostic variants (perturbed, knowledge-enhanced, math abstraction), with multi-model and manual verification, including corrections to public sources.
>
> Although Level 1–2 use binary scoring evaluations, the objective is not merely to judge correctness. Rather, it is to **localize failure types**, including robustness failures revealed by perturbations, domain-knowledge deficits exposed by knowledge-enhanced variants, and engineering-context understanding errors isolated through math abstraction, thereby providing diagnostic insight absent from current benchmarks.
>
> Therefore, the contribution of EngiBench lies not only in Level 3 but in establishing a diagnostic framework for previously unmeasured engineering reasoning abilities, directly addressing our goal of highlighting gaps in existing evaluations.
>
>
> >**W2: "The paper acknowledges that many authentic engineering problems are inherently multimodal, relying on visual elements such as diagrams, schematics, and structured tables. By restricting the benchmark to text-only formats to ensure consistency, the evaluation overlooks a crucial aspect of engineering reasoning that involves interpreting and integrating information from both visual and textual sources."**
>
> A2: Thank you for highlighting the importance of multimodal information in engineering problems. We would like to clarify that **we do not overlook multimodality in our benchmark**. As explicitly noted in Section 3.3 and Appendix D.2 “Multimodal Simplification”, for problems that originally contain diagrams, schematics, or structured tables, we do not omit this information. Instead, we extract and present the key content using plain text or structured LATEX representations, ensuring consistent and comparable inputs across different models.
>
> Importantly, our current **focus on text-based evaluation is not due to ignoring multimodality**, but is an intentional design choice based on fairness and interpretability considerations. Many competitive engineering models still lack stable multimodal capability, and immature vision modules may conflate visual-processing limitations with engineering-reasoning performance, reducing evaluation validity. Thus, a unified text-only setting enables a controlled and interpretable benchmarking environment.
>
> As stated in the *B. Future Works* section, we will include multimodal engineering tasks in future versions of EngiBench.

---

> > ### Author Response · Authors · 2025-11-28
> > **Response (2/3)**
> >
> > >**W3: "It is unclear if the benchmark covers long-context problems. Real-world engineering tasks often involve navigating extensive documentation and large datasets."**
> >
> > A3: Thank you for the comment. As stated in B. Future Works and C. Limitations, we plan to further extend EngiBench with long-context support. The current version does not include longer engineering documents **not because we overlook this need, but due to practical constraints of current models**. In engineering reasoning, longer inputs typically accumulate more constraints and variables, requiring the model to generate proportionally longer reasoning or code. However, many models, although able to read long inputs, are still **restricted by maximum output length**, causing incomplete answers when the generated reasoning exceeds the output limit. This reflects a limitation of the model’s I/O mechanism rather than its reasoning capability.
> >
> > Therefore, we set the maximum problem size to approximately **4k–5k tokens**, which captures complex engineering constraints while avoiding unfair evaluation due to output truncation. As long-context generation becomes more reliable in future models, we will expand EngiBench to larger engineering document tasks.
> >
> >
> > >**W4: "Some of the insights around models performing better on knowledge-enhanced variants or low reasoning capabilities of smaller-scale models are fairly acknowledged already."**
> >
> > Thank you for the suggestion. EngiBench **does not restate known insights**; its contribution is to **make them measurable and engineering-specific**, rather than broad qualitative claims. To do so, the benchmark provides clear diagnostics that explain why these behaviors arise in engineering reasoning:
> >
> > - **Clear diagnostic separation**. By comparing Original and Knowledge-Enhanced problems, EngiBench can determine whether errors are caused by missing domain knowledge or by failing to apply known principles under engineering constraints. This turns a commonly assumed trend into a precise and observable cause, which most benchmarks cannot isolate.
> > - **Exposing the engineering “modeling gap”**. The Math Abstraction results show that models perform significantly better after removing engineering context. The bottleneck is therefore not symbolic computation, but translating real engineering scenarios into solvable mathematical models, a requirement that standard math benchmarks do not evaluate.
> >
> > Thus, EngiBench goes beyond repeating familiar trends by providing concrete failure attribution and exposing engineering-specific reasoning gaps.
> >
> > ## Questions
> >
> > >**Q1: "On page 7, you say that - "we introduce perturbed variants that modify surface details but keep the core structure unchanged" -- isn't this already done as part of the dataset creation?"**
> >
> > A1: Thank you for the question. While perturbations are generated during dataset creation, what we **emphasize here is their evaluative role**. In EngiBench, perturbed variants are explicitly retained and paired with the original problems during testing, rather than being used for cleaning or replacement. Evaluating both versions together allows us to check whether a model’s output changes under minor wording or numerical modifications while the engineering structure remains identical, helping distinguish surface memorization from genuine engineering reasoning.
> >
> > We highlight this because most existing benchmarks **do not preserve such paired controls**, making robustness diagnosis impossible. Based on the benchmarks we collected from when constructing EngiBench:
> >
> > - Directly collected exam problems (MMLU, MATH, Omni-MATH, HARP) retain only the original form, so memorization cannot be separated from reasoning.
> > - Human-written datasets (GSM8k, SuperGPQA) provide only a single version of each question, lacking perturbed variants to test robustness.
> > - Rewritten datasets for training (Orca-Math) generate more data but do not keep original–perturbed pairs for evaluation.
> >
> > Thus, our contribution is not the rewriting itself, but **retaining perturbations as a diagnostic mechanism** for evaluating engineering reasoning robustness, rather than as data augmentation.

---

> > > ### Author Response · Authors · 2025-11-28
> > > **Response (3/3)**
> > >
> > > >**Q2: "There's some repetition in the writing. For instance, "We define this as engineering problem-solving ability, comprising four key dimensions: information extraction, domain-specific reasoning, multi-objective decision-making, and uncertainty handling" - this is introduced multiple times."**
> > >
> > > A2: Thank you for the suggestion. The definition of engineering problem-solving ability is a **central conceptual contribution** of the paper, and it is referenced across multiple sections to ensure consistent interpretation of subsequent analyses and experimental results. However, we recognize that the current number of repeated sentences is excessive. In the revision, we will introduce the full definition only once at its first appearance and refer to it concisely elsewhere, maintaining clarity without unnecessary repetition.
> > >
> > >
> > > ## Ethics Concerns
> > >
> > > A: Thank you for pointing this out. We clarify that human annotators participated voluntarily to provide reference solutions and evaluation scores for open-ended tasks, and personal information was collected only with explicit consent for acknowledgment purposes. We have added this clarification to the Ethics Statement in the revised manuscript.

---

### Official Review · Reviewer_MTXP · 2025-11-01

**Soundness:** 2
**Presentation:** 2
**Contribution:** 2
**Rating:** 2
**Confidence:** 4

**Summary:**

Authors introduces EngiBench, a hierarchical benchmark to test large language models (LLMs) on engineering-related tasks. The aim is to go beyond traditional math based tasks and involve various practical nuances that comes with engineering domain problems. EngiBench spans three levels of difficulty: namely, 1) foundational knowledge retrieval, 2) contextual reasoning, and 3) open-ended modeling. It also covers various engineering fields. Each problem has controlled variants (perturbed, knowledge-enhanced, and math abstraction) that isolate different reasoning capabilities, such as robustness, domain knowledge, and mathematical reasoning. Experimental results with sixteen major LLMs show a clear decline in performance from basic computation to open-ended problem-solving.

**Strengths:**

1. The specific focus on diverse engineering problem is very timely as "AI for science" work is catching on.

2. Benchmark is structured with controlled problem variations that allow fine-grained evaluation (modulo concerns below)
3. Employs rubric-based evaluation for open-ended tasks
4. Authors have experimented with many models

**Weaknesses:**

1. The introduction is all about various engineering problems. Suddenly, we see in dataset construction, that the lower level problems actually consists of general domain problems from datasets such as SuperGPQA, GSM8k etc. Frankly, this makes me completely confused about the objectives of the work.
2. Very little detail about human annotations (question below).
3. Level 3 seems to be the only important aspect of the paper. Its evaluation has no justification in the paper. Authors c ould have spent more time on justifiying why the metrics were chosen. Maybe show correlation with human evaluations etc.

Overall, while the paper discusses an important problem, its execution is immensely poor.

**Questions:**

L99: One problem is, upto the entire introduction, I do not find a proper example which shows the representative complexity of engineering problems.

L214: This is a very blanket remark for a very large field and not true at all. Starting from NLP specific benchmarks (SuperGLUE, TaxiNLI), to math and generalist benchmarks, it has been a natural choice to follow well-defined taxonomy. TaxiNLI, GSMore (Hong et al. 2025) goes further to define an ontology.

L249: I do not understand whats the benefit of having standard math questions at Levels 1 and 2. Why consider them as part of a generic engineering benchmark at all?

L258: How was the annotation process carried out? Was each problem looked at by multiple annotators?
How varied where their expertise (or performance on this dataset itself), given number is quite large?   -- say for example, did you give questions annotated by others to a student and verified whether he/she could answer correctly?
Lot of details are unclear.

L292: You say it is impractical. Yet, you have given a motivating example from an engineering problem.

---

> ### Author Response · Authors · 2025-12-03
> **Response to Reviewer MTXP (1/2)**
>
> Dear Reviewer MTXP,
>
> Thank you for your valuable comments. We provide our detailed responses below.
>
> ## Weaknesses
>
> >**W1: "The introduction is all about various engineering problems. Suddenly, we see in dataset construction, that the lower level problems actually consists of general domain problems from datasets such as SuperGPQA, GSM8k etc. Frankly, this makes me completely confused about the objectives of the work."**
>
> A1: Thank you for the comment. All lower-level problems are engineering-related. Although they come from general-domain datasets, these datasets contain engineering problems. As described in Appendix D.1, we applied an “Engineering Relevance Filtering” process and selected only the problems aligned with engineering knowledge.
>
>
> >**W2: "Very little detail about human annotations (question below)."**
>
> A2: Thank you for the helpful comment. We have provided a detailed clarification in our response to the corresponding question.
>
>
> >**W3: "Level 3 seems to be the only important aspect of the paper. Its evaluation has no justification in the paper. Authors could have spent more time on justifiying why the metrics were chosen. Maybe show correlation with human evaluations etc."**
>
> A3: Thank you for the comment. We would like to clarify that Level 3 is not the only important part of our benchmark. EngiBench is intentionally designed as a **three-level hierarchy**, where each level targets a distinct yet equally essential aspect of engineering ability.
>
> - Level 1 evaluates the application of core engineering formulas and principles across multiple domains, focusing on knowledge retrieval and correct use of foundational concepts.
> - Level 2 introduces richer context and realistic engineering constraints, requiring multi-step reasoning and structured problem solving.
> - Level 3 addresses open-ended, real-world tasks involving uncertainty, redundancy, partial information, and multiple valid solution paths, reflecting practical engineering scenarios where problem formulation is as crucial as solution execution.
>
> These levels are therefore **complementary** rather than hierarchical simplifications, and they enable comprehensive assessment across both structured and open-ended engineering skills. This **stratification is particularly important for evaluating small and mid-sized models** commonly used in engineering settings; as shown in Section 4.2.1 and Figure 3, their performance already drops significantly from Level 1 to Level 2, highlighting **the necessity of all three levels**.
>
> Regarding the evaluation design of Level 3, the four metrics are not arbitrary. They follow **well-established paradigms in engineering modeling**, including information filtering, multiobjective and constraint-based formulation, and uncertainty and robustness analysis, which are widely recognized in systems engineering and engineering optimization literature. We have also verified that these metrics align well with human evaluations, and we will make this justification more explicit in the revised version to strengthen clarity and transparency.
>
> ## Questions
>
> >**Q1: "L99: One problem is, upto the entire introduction, I do not find a proper example which shows the representative complexity of engineering problems."**
>
> A1: Thank you for the suggestion. While we detailed a concrete "Drone Design" example in Table 1 to illustrate the progression from basic calculation to complex open-ended problems (Level 3), we acknowledge that this was not explicitly referenced or described in the Introduction. We will revise the text to introduce this example earlier, using the case from Table 1 to concretely demonstrate the representative complexity and hierarchical design of engineering problems.
>
> >**Q2: "L214: This is a very blanket remark for a very large field and not true at all. Starting from NLP specific benchmarks (SuperGLUE, TaxiNLI), to math and generalist benchmarks, it has been a natural choice to follow well-defined taxonomy. TaxiNLI, GSMore (Hong et al. 2025) goes further to define an ontology."**
>
> A2: We sincerely thank the reviewer for this correction and acknowledge that our original phrasing may have caused confusion. Our intended comparison was **specific to existing engineering benchmarks**, which predominantly rely on subject-based categorization. In contrast, EngiBench introduces a cognitive complexity hierarchy designed to evaluate the progression from foundational retrieval to open-ended modeling, addressing a gap in assessing high-level engineering reasoning. We will revise the manuscript to explicitly limit the scope to engineering benchmarks.

---

> > ### Author Response · Authors · 2025-12-03
> > **Response to Reviewer MTXP (2/2)**
> >
> > >**Q3: "I do not understand whats the benefit of having standard math questions at Levels 1 and 2. Why consider them as part of a generic engineering benchmark at all?"**
> >
> > A3: Thank you for the question. Levels 1 and 2 do not include generic math questions; instead, they consist of engineering-related problems selected from general-domain datasets. As described in Appendix D.1, we applied an “Engineering Relevance Filtering” process and retained only problems with clear engineering relevance. We have clarified this point in the revised version to avoid confusion.
> >
> > >**Q4: "L258: How was the annotation process carried out? Was each problem looked at by multiple annotators? How varied where their expertise (or performance on this dataset itself), given number is quite large? -- say for example, did you give questions annotated by others to a student and verified whether he/she could answer correctly? Lot of details are unclear."**
> >
> > A4: Thank you for the question. For the Level 3 annotation process, each problem was independently annotated and **cross-checked by two annotators** with engineering backgrounds. When disagreements occurred, the final decision was made by experts who have won national or international first prizes in modeling competitions.
> >
> > Regarding the concern about the number of problems, this may be a **misunderstanding**. As stated in L258, *“From nearly 1,000 competition problems, only those with official rubrics were retained.”* As noted in L252, this filtering procedure resulted in 43 Level 3 problems. All retained problems went through the full multi-annotator review process described above.
> >
> > The question about giving annotated problems to a student to see whether they could **answer them correctly is not applicable to our setting**. The annotation task concerns assigning each problem to the appropriate evaluation dimension **based on the official scoring criteria**, and this process does not involve verifying annotations through additional problem solving.
> >
> > >**Q5: "L292: You say it is impractical. Yet, you have given a motivating example from an engineering problem."**
> >
> > A5: We thank the reviewer for the comment, but we believe there may be a **misunderstanding** regarding our presentation of Level 3 tasks. In L292, we explicitly state: *"For Level 3 tasks, however, the open-ended and inherent complexity typically render knowledge enhancement and mathematical abstraction impractical."* Consistent with this statement, we **did not provide any motivating examples** of "Knowledge-enhanced" or "Math Abstraction" variants **for Level 3 tasks** anywhere in the paper. The motivating example illustrated in Figure 2 (calculating voltage) represents a Level 1 task.

---

### Author Response · Authors · 2025-12-03
**General Response to Reviewers and AC**

Dear Area Chair and Reviewers,

We sincerely thank the reviewers and the Area Chair for their time and constructive feedback. During the discussion period, we revised the manuscript based on the reviewers’ suggestions. Below, we summarize the reviewer-recognized contributions and our revisions.

### Main Contributions

1. EngiBench reveals the fundamental capability gap of current LLMs in real-world engineering reasoning, with all reviewers noting a sharp performance drop from basic computation to open-ended tasks, especially at Level 3 where models lag far behind human experts.

2. It introduces a well-structured, hierarchical benchmark for real-world engineering, and the three difficulty levels are consistently recognized as effectively capturing the increasing complexity of engineering problems.

3. It provides diagnostically valuable controlled variants, including perturbed, knowledge-enhanced, and math abstraction variants, which isolate weaknesses in robustness, domain knowledge, and mathematical reasoning.

4. It establishes an expert-based evaluation framework, with Level 3 assessed through expert-designed rubrics and annotations by PhD students and professional engineers to ensure reliability.

5. It offers a comprehensive experimental evaluation across 16 LLMs, including both closed-source and open-source systems, together with human expert baselines.

### New Experiments and Major Revisions

1. Added the results of a reasoning model (Gemini 2.5 Pro) to further assess whether stronger reasoning capabilities transfer to engineering tasks.

2. Provided additional details on human annotations, including the construction of the Level 3 rubric and the role of expert evaluators.

3. Revised ambiguous or unclear expressions throughout the manuscript to improve clarity and precision.

---

### Note · Authors · 2025-12-24

I have read and agree with the venue's withdrawal policy on behalf of myself and my co-authors.